

Climate
of the Past

# The remote response of the South Asian Monsoon to reduced dust emissions and Sahara greening during the middle Holocene

**Francesco S. R. Pausata**[1], **Gabriele Messori**[2,3], **Jayoung Yun**[4], **Chetankumar A. Jalihal**[5,6], **Massimo A. Bollasina**[4], and **Thomas M. Marchitto**[7]

[1]Centre ESCER (Étude et la Simulation du Climat à l'Échelle Régionale) and GEOTOP (Research Center on the dynamics of the Earth System), Department of Earth and Atmospheric Sciences, University of Quebec in Montreal, Montreal, Canada [TS1]
[2]Department of Earth Sciences, Uppsala University, and Centre of Natural Hazards and Disaster Science (CNDS), Uppsala, Sweden
[3]Department of Meteorology, Stockholm University, and Bolin Centre for Climate Research, Stockholm, Sweden
[4]School of GeoSciences, University of Edinburgh, Edinburgh, UK
[5]Centre for Atmospheric and Oceanic Sciences, Indian Institute of Science, Bengaluru, India
[6]Divecha Centre for Climate Change, Indian Institute of Science, Bengaluru, India
[7]Department of Geological Sciences and Institute of Arctic and Alpine Research, University of Colorado Boulder, Boulder, CO, USA

**Correspondence:** Francesco S. R. Pausata (pausata.francesco@uqam.ca)

**Abstract.** Previous studies based on multiple paleoclimate archives suggested a prominent intensification of the South Asian Monsoon (SAM) during the mid-Holocene (MH, $\sim$ 6000 years before present). The main forcing that contributed to this intensification is related to changes in the Earth's orbital parameters. Nonetheless, other key factors likely played important roles, including remote changes in vegetation cover and airborne dust emission. In particular, northern Africa also experienced much wetter conditions and a more mesic landscape than today during the MH (the so-called African Humid Period), leading to a large decrease in airborne dust globally. However, most modeling studies investigating the SAM changes during the Holocene overlooked the potential impacts of the vegetation and dust emission changes that took place over northern Africa. Here, we use a set of simulations for the MH climate, in which vegetation over the Sahara and reduced dust concentrations are considered. Our results show that SAM rainfall is strongly affected by Saharan vegetation and dust concentrations, with a large increase in particular over northwestern India and a lengthening of the monsoon season. We propose that this remote influence is mediated by anomalies in Indian Ocean sea surface temperatures and may have shaped the evolution of the SAM during the termination of the African Humid Period.

## 1 Introduction

The South Asian Monsoon (SAM) directly affects the climate of the Indian subcontinent and indirectly influences far-afield regions through atmospheric and oceanic teleconnections (e.g., Lau, 1992; Liu et al., 2004). Due to its key role for regional and global hydrological cycles, much attention has been devoted to better understand and predict its variability on multiple timescales, including its long-term future changes (e.g., Huo and Peltier, 2020; Swapna et al., 2018). However, SAM future projections are highly uncertain (e.g., Huang et al., 2020), and even representing recent trends and identifying their drivers has been challenging (e.g., Mishra et al., 2018) due to the relatively short modern observational record that spans roughly a century. Hence, investigating past SAM changes is of utmost importance to better understand its dynamics and future evolution.

Dramatic shifts in the intensity of the SAM occurred at the end of the deglaciation (Bird et al., 2014; Campo et al., 1982; Dallmeyer et al., 2013; Fleitmann et al., 2003; Gill et al., 2017; Saraswat et al., 2013) when stronger boreal summer insolation, increasing greenhouse gas concentrations, and shrinking ice sheets triggered a strengthening of the Northern Hemisphere summer monsoon systems (Jalihal et al., 2019a; Sun et al., 2019). In particular, the increased orbital forcing enhanced moisture transport from the Indian Ocean to the Indian subcontinent, leading to stronger monsoonal precipitation there (e.g., Dallmeyer et al., 2013; Texier et al., 2000). These changes occurred in parallel with a prolonged period of intense precipitation over northwestern Africa – labeled the African Humid Period. The African Humid Period spanned the entire early and middle Holocene (15 000–4000 BP TS2) and had far-reaching local and global climatic influences (Muschitiello et al., 2015; Pausata et al., 2017a, b; Piao et al., 2020; Sun et al., 2019). Locally, it coincided with a major intensification of the West African Monsoon (WAM) and a greening of the present-day Sahara. Amongst its many remote impacts, the WAM strengthening contributed to the greening of the arid and semi-arid regions of east and south Asia (see Pausata et al., 2020, for a recent review). Indeed, the large circulation changes instigated by the African Humid Period greening of the Sahara, together with the associated changes in sea surface temperatures, have likely complemented orbital changes in modulating the SAM (see Texier et al., 2000).

Numerical paleoclimate simulations have typically been deficient in capturing the dramatic shift in the WAM in the mid-Holocene, even when changes in orbital forcing and land-surface cover were considered (Harrison et al., 2014). A crucial factor that has been largely overlooked until recently has been the role played by the sharp decrease in Saharan dust emissions, which occurred in conjunction with the greening (Arbuszewski et al., 2013; McGee et al., 2013). Pausata et al. (2016), Gaetani et al. (2017), and Messori et al. (2019) have shown that atmospheric dust loading profoundly affects monsoonal dynamics and, while changes in vegetation do lead to increased monsoonal precipitation, a better agreement with proxy data is only reached when a dust reduction is also simulated (see Tierney et al., 2017). The latter strengthens the effects of land-surface changes, leading to a further increase and northward extension of the WAM. Egerer et al. (2018) have also shown that accounting for both vegetation and dust feedbacks leads to a better match between model simulations and paleoclimate reconstructions from the northwestern African margin. Another recent study (Thompson et al., 2019) has suggested a contribution from dust aerosol reduction of about 15 %–20 % to the total rainfall over the Sahara; however, they also revealed that dust–cloud interactions have the opposite effect compared to the direct radiative effect on rainfall in northern Africa during the mid-Holocene (MH) CE1. Hopcroft and Valdes (2019) showed the dependence on the modeled dust optical properties and particle size range of the impacts on WAM rainfall, leading to potential overestimation of the direct radiative effect on precipitation.

Through a set of sensitivity experiments performed with an Earth system model, Pausata et al. (2017a, b) have shown that the strengthening of the WAM and the associated vegetation and dust feedbacks during the MH are able to affect the El Niño–Southern Oscillation variability as well as tropical storm activity worldwide. Using the same set of simulations, Piao et al. (2020) showed that a vegetated Sahara leads to an enhancement of the western Pacific subtropical high, which in turn strengthens the east Asian summer monsoon. Sun et al. (2019) highlighted that Northern Hemisphere land monsoon precipitation significantly increases by over 30 % under the effect of the "Green Sahara". However, a systematic evaluation of the joint impacts of atmospheric dust loading reductions, Saharan land-cover changes, and insolation changes on the SAM during the mid-Holocene is lacking in current literature.

Here, we address this gap with the aim of providing insights into future SAM changes. Indeed, a number of recent studies have projected future increases in Sahelian precipitation (Biasutti, 2013; Giannini and Kaplan, 2019) associated with a surface greening and reduced dust emissions (Evan et al., 2016).

The remainder of the paper is organized as follows: the climate model used and the experimental design are described in Sect. 2. Next, we examine SAM changes during the summer, both at the surface and aloft (Sect. 3). A discussion and conclusions follow in Sect. 4.

## 2   Model description and experimental design

The study is based on a set of simulations performed with an Earth system model (EC-Earth version 3.1). EC-Earth version 2 participated in the fifth phase of the Coupled Model Intercomparison Project (CMIP5) and version 3 will participate in CMIP6 (http://www.ec-earth.org/, last access: 1 June 2021 TS3). The model is comprised of the Integrated Forecasting System (IFS cycle 36r4) for the atmosphere, the Nucleus for European Modelling of the Ocean version 2 (NEMO2) for the ocean, and the Louvain-la-Neuve sea-ice Model version 3 (LIM3) for sea ice (Hazeleger et al., 2010; Yepes-Arbós et al., 2016). The IFS model includes the Tiled ECMWF Scheme for Surface Exchanges over Land incorporating land surface hydrology (H-TESSEL) and is run at T159 horizontal spectral resolution corresponding to roughly 1.125° in longitude and latitude, with 62 vertical levels. NEMO has a 1° horizontal resolution except at the Equator, where it increases to 1/3° (Sterl et al., 2012), and 46 vertical levels. The different components are coupled via the OASIS3 coupler. Relevant for this study, vegetation cover and monthly aerosol concentrations (Tegen et al., 1997) are prescribed in the model; however, the indirect effect of aerosols

on clouds is not considered. A detailed description of the aerosol components can be found in Hess et al. (1998). The main characteristics of dust particles are reported in Table A1.

We analyze a MH experiment ($MH_{PMIP}$), which follows the protocol for the standard mid-Holocene simulations in accordance with the third phase of the Paleoclimate Modelling Intercomparison Project (PMIP3) (Taylor et al., 2009, 2012) and three sensitivity experiments performed by Pausata et al. (2016) and Gaetani et al. (2017) (Table 1). The $MH_{PMIP}$ includes mid-Holocene orbital forcing and greenhouse gas concentrations, pre-industrial land cover, and airborne dust concentrations. The three sensitivity experiments were carried out to investigate the effects of changes to land-cover conditions and dust concentration in isolation as well as in combination. In the $MH_{GS}$ ("Green Sahara") setup, the vegetation type (and related parameters; see below) over the Sahara (defined as the area within 11–33° N, 15° W–35° E) is prescribed to be evergreen shrub, representing an idealized African Humid Period scenario, while dust concentration is left unaltered at its pre-industrial (PI) amounts. In the $MH_{RD}$ ("reduced dust") setup, the dust concentration over northern Africa is reduced by up to 80 % relative to pre-industrial values (see Figs. 1 and S1 in Gaetani et al., 2017). Outside northern Africa, dust concentrations smoothly transition to pre-industrial values. Over India and the Arabian Sea the reduction of dust concentrations ranges between 20 % (eastern Indian subcontinent) and 60 % (Horn of Africa and the Middle East); for more details, see Fig. S1 in Pausata et al. (2016). In the $MH_{RD}$ experiment the land-surface properties are kept to PI values. The final experiment ($MH_{GS+RD}$) considers the case where both vegetation and dust changes described above are simultaneously prescribed. The imposed changes in vegetation type correspond to important changes in surface albedo and leaf area index (LAI) as summarized in Table 2. Both albedo and LAI are fixed throughout the simulations and are not modulated by modeled processes. Under the Green Sahara scenario, the albedo decreases from 0.3 to 0.15, while the LAI increases from 0.2 to 2.6. For a more detailed description of these simulations, the reader is referred to Pausata et al. (2016) and Gaetani et al. (2017). These sensitivity experiments are compared to a PI simulation to investigate the role of each forcing in altering the SAM. The analysis focuses on the June–September (JJAS) period using the last 50 years of each experiment. Finally, the statistical significance of the differences between experiments at the 5 % level is evaluated by a two-tailed Student's $t$ test.

# 3   Results

This section discusses the SAM response in terms of local (Sect. 3.1) and large-scale changes (Sect. 3.2) to each forcing independently and together: orbital ($MH_{PMIP}$), orbital forcing and Sahara greening ($MH_{GS}$), orbital forcing and dust reduction ($MH_{RD}$), and orbital forcing, Sahara greening, and dust reduction ($MH_{GS+RD}$). In Sect. 3.3 we then compare the model findings to paleoclimate archives.

## 3.1   Changes in surface climate

### 3.1.1   Precipitation

In the PI experiment, the SAM displays the most intense summertime (particularly June and July) precipitation over the west coast of the Indian subcontinent and the Himalayan foothills (Fig. 1a), in overall agreement with observations (Figs. A1 and A2). The $MH_{PMIP}$ experiment simulates a general increase in SAM rainfall over south Asia compared to PI (Fig. 2a) as also shown by other PMIP model experiments (e.g., Zhao and Harrison, 2012), particularly over southern India and the Himalayan foothills. In contrast, decreased precipitation is seen over most of the Bay of Bengal, South China Sea, and Thailand. This decrease in precipitation is a result of the reduced surface latent heat flux over the ocean as shown in Jalihal et al. (2019b). This results in a decrease in the net energy flux into the atmosphere over these regions, leading to a decline in precipitation. A precipitation anomaly dipole is simulated along the equatorial Indian Ocean, with increased precipitation to the west and decrease to the east. The greening of the Sahara ($MH_{GS}$) leads to a general intensification of the anomaly pattern simulated when only including orbital forcing ($MH_{PMIP}$; Fig. A3a). However, some peculiar characteristics emerge: in particular, the precipitation increases over a broad swathe of northwestern India and Pakistan, while it decreases over the Western Ghats (cf. CE2 Fig. 2a and b, and see also Fig. A3a). The positive rainfall anomaly over the western equatorial Indian Ocean extends eastward, strongly reducing the negative precipitation anomaly in the eastern side of the basin. The reduction in precipitation over the Bay of Bengal, southeast Asia, and Thailand further intensifies. The reduced Saharan dust ($MH_{RD}$) leads to a pattern that is very similar – albeit with weaker anomalies – to the orbital-only forcing ($MH_{PMIP}$; Fig. A3b); however, the precipitation increase over southern India is confined to east of the Western Ghats, while a small decrease in rainfall is simulated along the western coast of the Indian subcontinent (cf. Fig. 2a and c, and see also Fig. A3b). When combining vegetation and reduced dust ($MH_{GS+RD}$), features of both simulations ($MH_{GS}$ and $MH_{RD}$) are preserved (Fig. 2d): the $MH_{GS+RD}$ anomaly pattern in the region is almost exactly the linear combination of the $MH_{GS}$ and $MH_{RD}$ experiments (Fig. A4). For example, the reduced precipitation over the Western Ghats is further enhanced in the $MH_{GS+RD}$, while the increase over the Himalayan foothills is reduced compared to the $MH_{GS}$, which is due to the effect of the dust reduction (cf. Fig. 2a and c, and see also Fig. A3b).

While seasonal-mean precipitation determines the overall amount of water supplied, subseasonal changes in the monsoon, such as a shift in the onset and/or the withdrawal, are

https://doi.org/10.5194/cp-17-1-2021

**Table 1.** Boundary conditions for all MH experiments.

| Simulation | Orbital forcing BP | GHGs | Saharan vegetation | Saharan dust |
|---|---|---|---|---|
| MH$_{PMIP}$ | 6000 | MH | Desert | PI |
| MH$_{GS}$ | 6000 | MH | Shrub | PI |
| MH$_{RD}$ | 6000 | MH | Desert | Reduced |
| MH$_{GS+RD}$ | 6000 | MH | Shrub | Reduced |

**Table 2.** Albedo and leaf area index (LAI) for desert, evergreen shrub, and the domain over which the vegetation changes are applied in each setup.

| | Vegetation type | Albedo | LAI | Domain |
|---|---|---|---|---|
| PS | Mainly desert | 0.30 | 0.18 | 11–33° N, 15° W–35° E |
| GS | Evergreen shrub | 0.15 | 2.6 | 11–33° N, 15° W–35° E |

key to determining the length and hence the precipitation rate over the monsoonal season. The SAM in the PI simulation starts in late May (Figs. 3 and A2a), with the monsoon then developing until early August and retreating in early September (Figs. 3 and A2a). In the MH$_{PMIP}$ experiment, the model simulates a delayed onset south of 15° N but not at higher latitudes (Fig. 3a). The withdrawal is, however, delayed at all latitudes, lengthening the overall duration of the monsoon by about 1 month. The Sahara greening (MH$_{GS}$) leads to a further lengthening of the monsoon season from April to October, and increased cumulative precipitation over a large part of the SAM region (Fig. 3b). The delayed onset is confined to the region well south of 10° N. While showing a lengthening of the monsoon season, the regions around the 15° N latitude band show a decrease in precipitation between June and mid-August (Fig. 3b). Dust reduction (MH$_{RD}$) leads to a much stronger delay of the monsoon onset that extends up to 25° N compared to the MH$_{PMIP}$ experiment (Fig. 3c). The withdrawal of the monsoon is also delayed and resembles the MH$_{PMIP}$ simulation (Fig. 3c). In the MH$_{GS+RD}$ case, the distribution of rainfall is dominated by the Sahara greening, but the footprint of the dust reduction is visible at the lower latitudes. These display a stronger decrease in precipitation than the MH$_{GS}$ simulation, in particular during the core monsoonal season – June to late August (Fig. 3d). Therefore, reduced dust seems to primarily reduce the south Asian monsoonal precipitation at low latitudes, while the greening of the Sahara increases the precipitation further north.

### 3.1.2 Surface temperature

The highest PI surface temperatures on the Indian subcontinent are simulated over its northwestern part, in agreement with observations (Fig. A1). However, while the temperature pattern is similar to observations, our simulation displays a cold anomaly over a large part of the domain (Fig. A1). The changes in the orbital forcing (MH$_{PMIP}$) do not remarkably alter the summer surface temperature over India and southeast Asia, with only a modest increase over central eastern India and up to 1 °C warming in southeast Asia (Fig. 4a). A large increase (even more than 3 °C) is instead simulated outside the area of direct influence of the SAM and in particular over the arid and semi-arid regions of south Asia and the Arabian Peninsula (Fig. 3a). A positive Indian Ocean dipole (IOD)-like pattern develops in the Indian Ocean with warmer sea surface temperatures (SSTs) of about 0.5–1 °C over the eastern equatorial Indian Ocean up to roughly 15° N and colder anomalies of up to 1.5 °C off the east coast of Indonesia (Fig. 4a). Colder SSTs are instead present over the northernmost part of the Arabian Sea, and warmer SSTs are prevalent over most of the Bay of Bengal, in contrast with the conventional IOD as seen in reanalysis data (e.g., Saji et al., 1999; Webster et al., 1999). The positive IOD-like pattern that develops under MH orbital forcing is responsible for some of the rainfall anomalies discussed above (Fig. 2a): in particular, the precipitation dipole along the Equator and increased rainfall over the southern tip of India (Fig. 2a), which are typical of a positive IOD pattern (Fig. A5). There are, however, differences in the anomalies over India and the Bay of Bengal in the simulations since the orbital forcing primarily drives these anomalies. A classical positive IOD leads to an increase in precipitation over the core monsoon zone and the northern Bay of Bengal and a decrease in precipitation over the southern Bay of Bengal, peninsular India, and the Himalayan foothills (see Ashok et al., 2001; Saji et al., 1999). The orbital forcing leads to a different response over land and the ocean (Fig. 2a).

The Sahara greening (MH$_{GS}$) leads to a similar anomaly pattern to that of MH$_{PMIP}$ in the surface temperature (cf. Fig. 4a and b, and see also Fig. A6a). However, the warming is more pronounced than in MH$_{PMIP}$ over the bulk of the domain, with northwestern India being an exception (Fig. A6a). This may be linked to the simulated increase in rainfall in

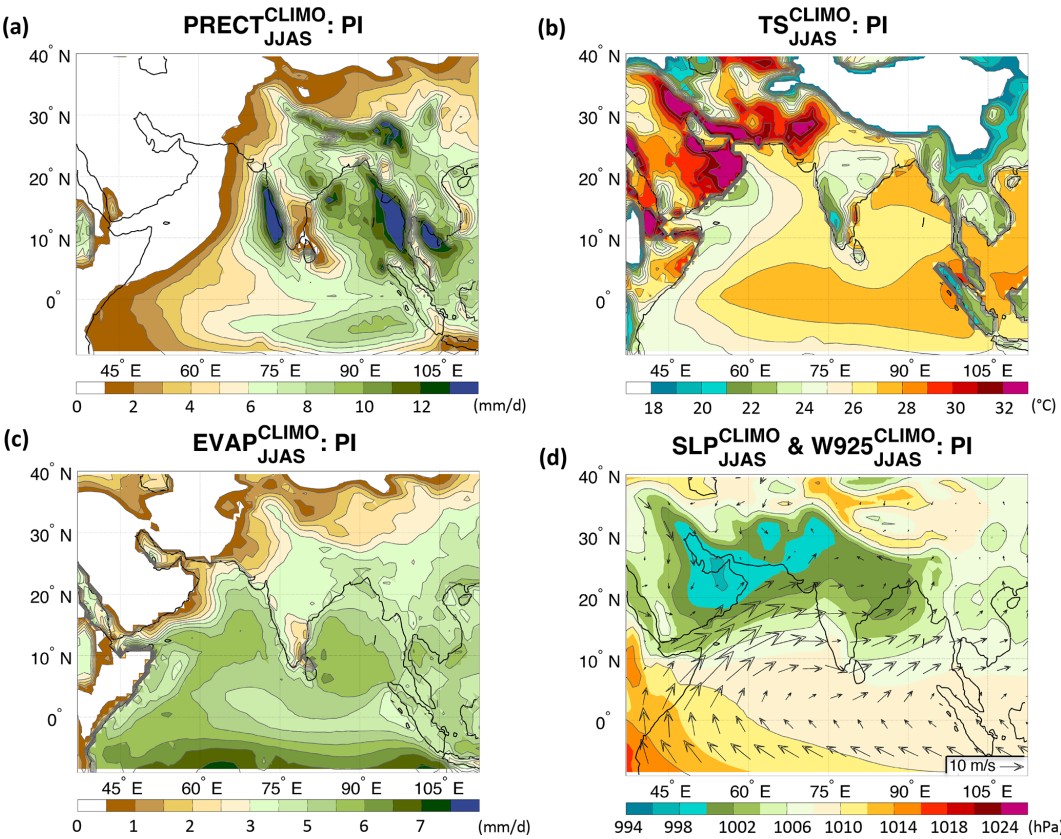

**Figure 1. (a)** Climatological summer (JJAS) precipitation (PRECT, mm/d); **(b)** surface temperature (TS, °C); **(c)** evaporation (EVAP, mm/d); and **(d)** sea level pressure (shadings, SLP, hPa) and 925 hPa wind (arrows, W925, m/s) for the PI experiment. The contour lines follow the color-bar scale.

the region (Figs. 2b, A3a). The cold SST anomalies over the northern Arabian Sea are replaced by warm anomalies that encompass almost the entire Indian Ocean north of the Equator. The temperature increases off the coast of the Somali peninsula and the southern tip of India exceed 1 °C. The positive IOD-like pattern is still present, particularly when considering relative anomalies as the SSTs over the equatorial Indian Ocean are generally warmer compared to the MH$_{PMIP}$. The IOD index is inversely related to the Indian monsoon rainfall (Fig. A7), thus suggesting that the MH, GS, and RD forcings have a dominant effect on the Indian monsoon. Reduced Saharan dust (MH$_{RD}$) leads to a widespread surface warming over the Arabian Peninsula, the Arabian Sea, and the Indian subcontinent (cf. Fig. 4a and c, and see also Fig. A6b). Such warming is partially due to a reduction in rainfall (Fig. A3b) and hence cloud cover in particular over southern India and southern Arabic Peninsula. Furthermore, while the reduced dust layer leads to a decrease in temperature in the mid-troposphere as dust is moderate-to-highly absorbing (single scattering albedo $\omega_0 < 0.95$; see Table A1), it increases the incoming solar radiation reaching the surface and hence favors surface warming. For the same reason, the cold SST anomalies in the northernmost Arabian

Sea in the MH$_{PMIP}$ experiment are replaced by a modest warm anomaly. Finally, the surface temperature response to the combined forcings (MH$_{GS+RD}$; Fig. 4d) closely resembles the linear combination of the two forcings (Fig. A4f), except over the regions facing the Gulf of Aden and southern Red Sea, where the cooling due to increased monsoonal precipitation in the MH$_{GS}$ (Fig. 2b) prevails over the warming associated with enhanced shortwave radiation in the MH$_{RD}$ (Fig. 5).

### 3.1.3 Evapotranspiration

From an impact-based perspective, changes in precipitation are only one part of the hydrological cycle, which also includes evaporation and, over land, transpiration as well. Therefore, it is important to also investigate the evapotranspiration changes during the MH climate to better understand the impacts of changes in orbital forcing and Saharan vegetation and dust on the water budget of south Asia. In the PI experiment, weak evapotranspiration is simulated over the dry subtropical desert regions, while rates in excess of 3 mm/d are present over the Indian subcontinent (Fig. 1c). In the MH$_{PMIP}$ experiment, the evapotranspiration is increased over

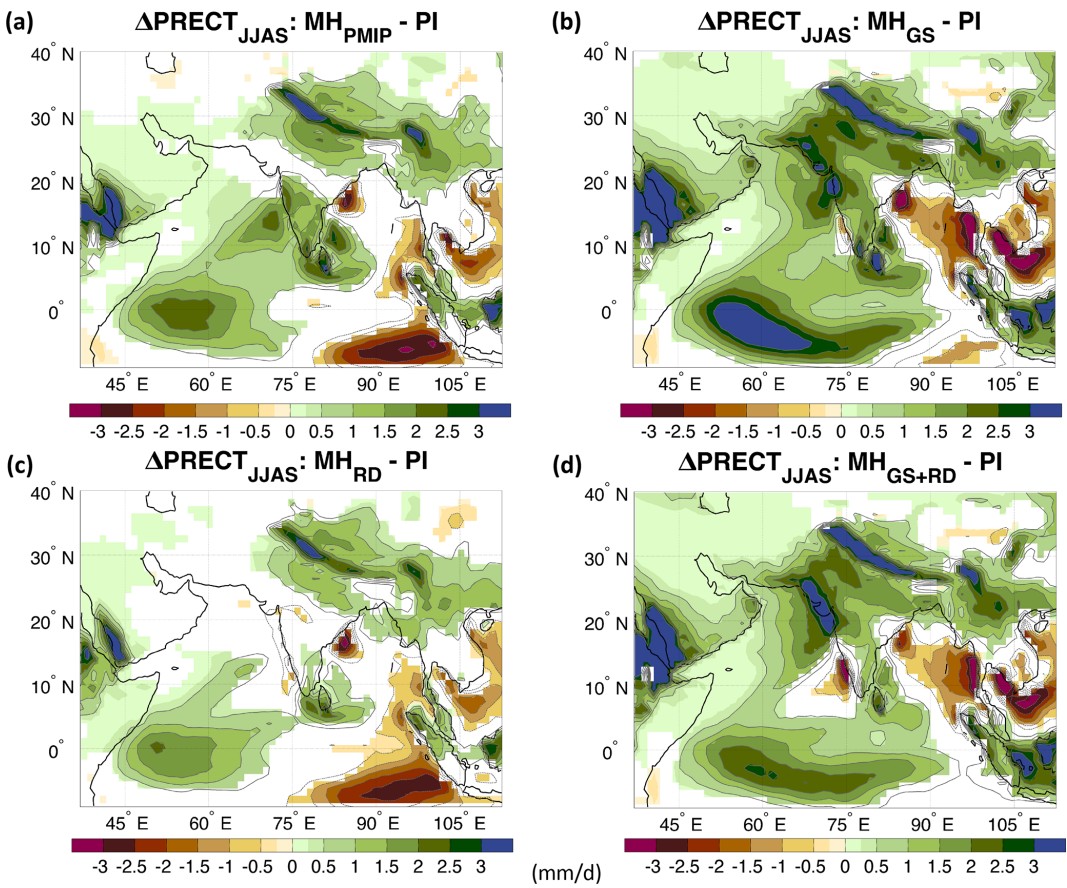

**Figure 2.** Changes in summer (JJAS) precipitation (PRECT; mm/d) for the **(a)** mid-Holocene-only orbital forcing ($MH_{PMIP}$); **(b)** the Sahara greening ($MH_{GS}$); **(c)** the dust-only reduction ($MH_{RD}$); and **(d)** the Sahara greening and dust reduction ($MH_{GS+RD}$) experiments relative to the PI reference simulation. The contour lines follow the color-bar scale (the 0 lines are omitted for clarity). Only differences significant at the 95 % confidence level using the Student $t$ test are shaded.

northwestern and southeastern India, the Tibetan Plateau, and southeast Asia (Fig. 6a). These regions are characterized by increases in precipitation and/or surface temperature (Figs. 2 and 4), which enhance the evapotranspiration. On the con-trary, the Indian Ocean displays a widespread decrease in evaporation rates except along the coast of Somalia. The Sahara greening ($MH_{GS}$) leads to a widespread increase in evapotranspiration across most of the Indian subcontinent, enhancing the anomaly pattern simulated in the $MH_{PMIP}$ ex-periment (Fig. 6a and b, and see also Fig. A8a). The reduc-tion in airborne dust ($MH_{RD}$) does not notably alter the evap-oration over land compared to the orbital-forcing-only exper-iment ($MH_{PMIP}$); however, it significantly increases the evap-oration over the Arabian Sea due to the increase in incoming solar radiation (panels a and c in Figs. 5 and 6, and A8b). Finally, the combined forcing ($MH_{GS+RD}$) leads to mainly positive anomalies over land, as in the $MH_{GS}$ case, while the effects of dust reduction dominate over the Arabian Sea and western Indian Ocean (Fig. 6d).

## 3.2 Changes in the large-scale monsoonal circulation

The PI sea level pressure (SLP) pattern displays a thermal low over the Arabian Peninsula extending into the northern part of the Indian subcontinent (Fig. 1d). This is associated with an anticyclonic circulation over the Indian Ocean lead-ing to a strong westerly flow across the Indian subcontinent and southeast Asia, which brings large amounts of moisture to these regions (Figs. 1a and d, A10a). The strong wester-lies over the Arabian Sea favor upwelling and explain the origin of the "cold pool" in that region (Fig. 1b). Anomalous easterlies along the equatorial Indian Ocean advect warmer water towards the western basin, leading to an increase in SSTs there. This further enhances convection over the west-ern equatorial Indian Ocean region. Concurrently, upwelling increases over the eastern equatorial Indian Ocean, and thus SSTs cool and precipitation decreases. As a result, a strong coupling between precipitation, circulation, and SST anoma-lies is established across the equatorial Indian Ocean, bearing close similarity with the pattern characteristic of the positive phase of the IOD. The subsequent changes in the low-level

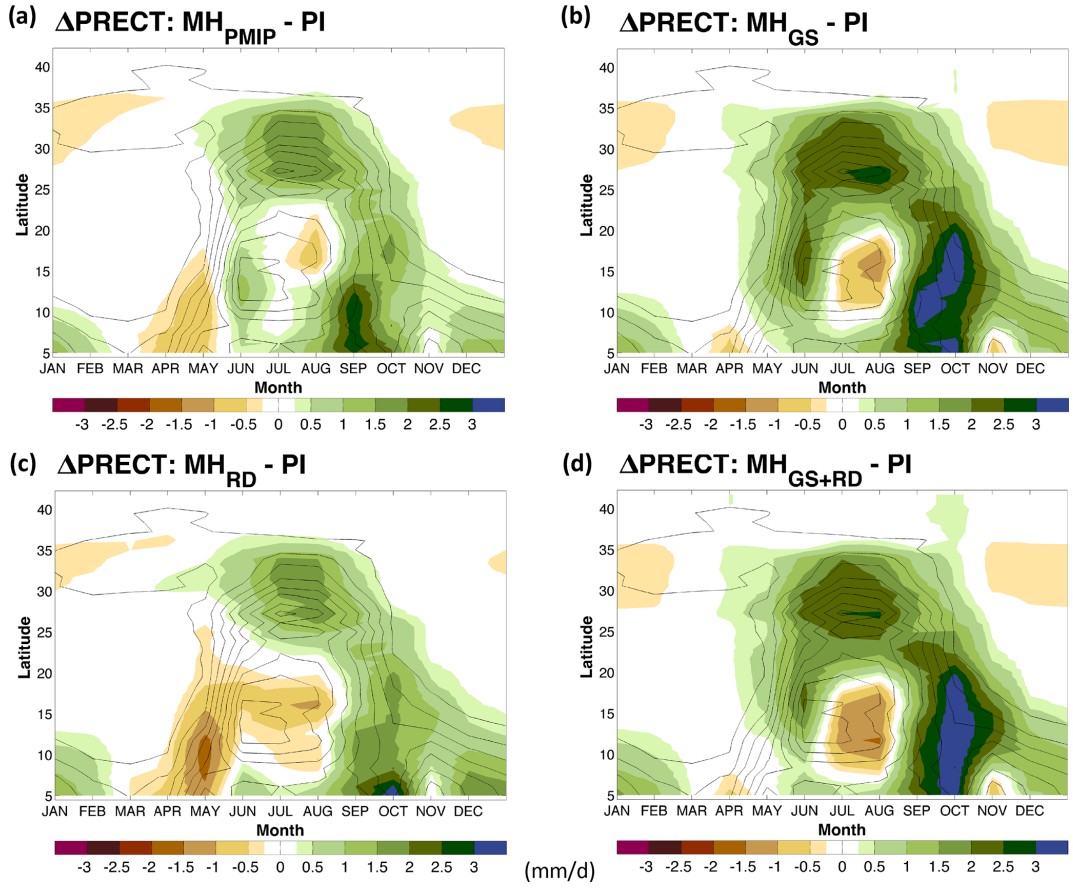

**Figure 3.** Changes in climatological seasonal cycle of zonal precipitation (PRECT; mm/d) between 65 and 95° E for **(a)** the mid-Holocene-only orbital forcing (MH$_{PMIP}$); **(b)** the Sahara greening (MH$_{GS}$); **(c)** the dust-only reduction (MH$_{RD}$); and **(d)** the Sahara greening and dust reduction (MH$_{GS+RD}$) experiments relative to the PI reference simulation. The contour lines show the climatological zonal precipitation of the PI experiment (1 mm/d intervals).

jet (intensification and a northward shift) lead to a cooling in the northern Arabian Sea (through enhanced coastal upwelling) and warming in the Bay of Bengal (through reduced winds and therefore evaporation). Thus, the SST anomalies in the Arabian Sea and the Bay of Bengal are different in the MH simulations from those for a positive IOD in the reanalysis data (e.g., Saji et al., 1999; Webster et al., 1999).

The MH orbital forcing (MH$_{PMIP}$) deepens the Saharan and Saudi Arabian heat low, while increasing the pressure over the Bay of Bengal relative to the PI. This anomaly pattern leads to an intensification of the easterly flow south of the Indian subcontinent, which then turns northeastward over the Arabian Sea (Fig. 7a), intensifying the monsoonal flow and in turn the upwelling in the region. The colder SSTs simulated over the northernmost part of the Arabian Sea are likely a direct consequence of this (Fig. 4a). The intensified monsoonal flow enhances the transport of moisture from the Bay of Bengal towards the western Indian Ocean and then the Arabian Sea and Indian subcontinent (Fig. 8a), explaining the rainfall changes seen in Fig. 2a. One may further connect the above circulation changes to the widespread decrease in evaporation rates simulated across most of the Indian Ocean and the concomitant increase along the coast of Somalia (Fig. 6a). For example, the latter evaporation increase is most likely driven by weakened monsoonal flow (Fig. 7a), which causes higher SSTs (Fig. 4a) and increases evaporation in the MH$_{PMIP}$ compared to the PI experiment (Fig. 6a). Conversely, the decreased evaporation in the northern Arabian Sea may be ascribed to the strengthened monsoonal flow, which increases upwelling and in turn cools the region (Fig. 4a). Finally, the weakened westerly flow around the southern tip of India may be responsible for decreased evaporation and a consequent increase in SSTs of that region. Under the Green Sahara conditions (MH$_{GS}$), the SLP anomaly pattern intensifies relative to the MH$_{PMIP}$ and shifts to the northwest, thus weakening the southwesterlies over the Arabian Sea, while strengthening the easterlies over the southern tip of India (Fig. 6a and b, and see also Fig. A9a). The latter anomaly can explain the decrease in precipitation over the western slopes of the Western Ghats and the

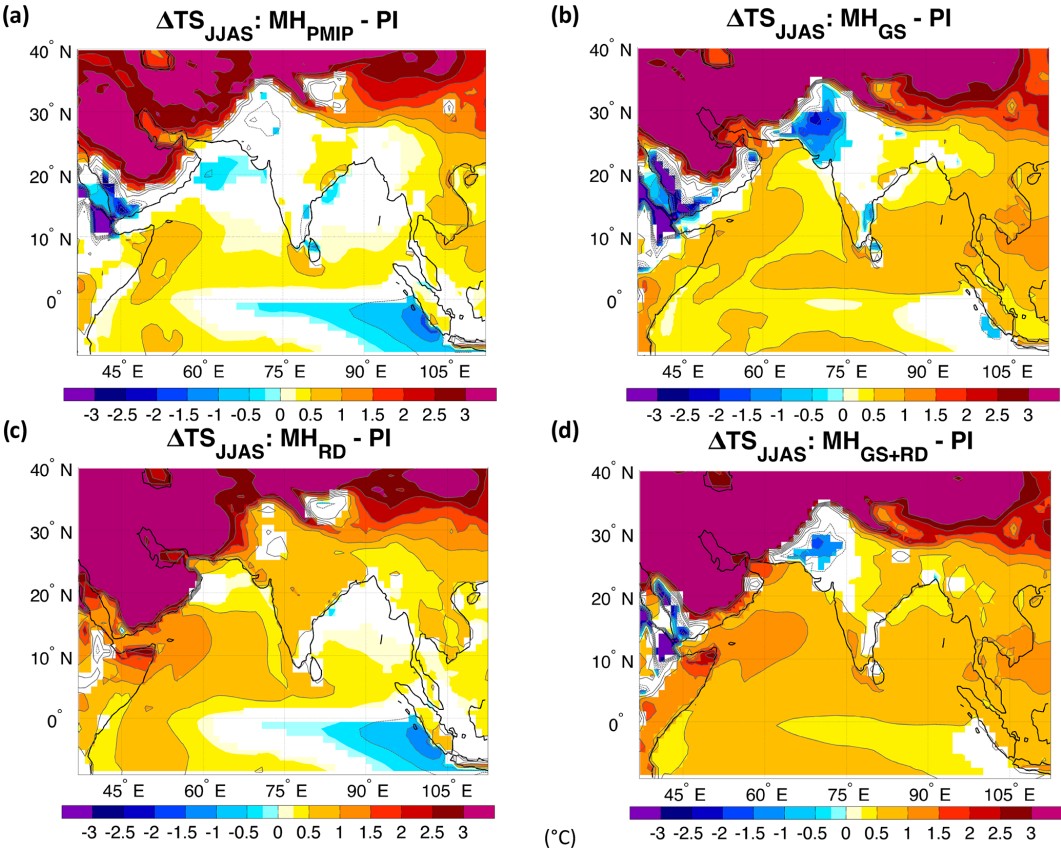

**Figure 4.** Changes in summer (JJAS) surface temperature (TS; °C) for the **(a)** mid-Holocene-only orbital forcing (MH$_{PMIP}$); **(b)** the Sahara greening (MH$_{GS}$); **(c)** the dust-only reduction (MH$_{RD}$); and **(d)** the Sahara greening and dust reduction (MH$_{GS+RD}$) experiments relative to the PI reference simulation. The contour lines follow the color-bar scale (the 0 lines are omitted for clarity). Only differences significant at the 95 % confidence level using the Student $t$ test are shaded.

increase on their eastern side. Although the southwesterly flow over the Arabian Sea is less intense than in the MH$_{PMIP}$ (Fig. A9a), the moisture advection is enhanced (Figs. 7b and A11a), which explains the increased precipitation and evapotranspiration over most of India (Figs. 2b, 6b, A8a). Indeed, the weakened atmospheric flow decreases the upwelling and in turn increases SSTs, favoring more evaporation over the Arabian Sea (Fig. A8a). Reduced Saharan dust (MH$_{RD}$) results in a northward expansion of the Mascarene High in the southern Indian Ocean and a weakening of the Saudi Arabian heat low relative to MH$_{PMIP}$ experiment (Fig. 6a and c, and see also Fig. A9b). This leads to a weakening of the Somali jet, a weaker coastal upwelling in the Arabian Sea favoring modest warm SST anomalies there (Fig. 4c), and ultimately a weaker moisture transport from the Arabian Sea to the southern half of the Indian subcontinent (Fig. 7a and c, and see also Fig. A11b). The weakened low-level winds relative to MH$_{PMIP}$ are consistent with the significant decrease in precipitation over western India (Fig. A3b). Further east, there is a strengthened northwesterly flow over the Bay of Bengal extending towards the western equatorial Pacific, associated with a decreased moisture convergence over

Bangladesh and northeastern India relative to the MH$_{PMIP}$ simulation (Fig. A11b). This circulation change causes a precipitation increase in the MH$_{RD}$ that is smaller than that in the MH$_{PMIP}$ relative to the PI (Figs. 2c and A3b). When combining both Sahara greening and dust reduction (MH$_{GS+RD}$), SLP anomalies are mostly a linear combination of the two forcings (Fig. 7d). In particular, the cyclonic footprint over the Indian Ocean and the easterly moisture transport from the Pacific to the Indian Ocean are both features of the MH$_{GS}$ experiment (Figs. A9 and A11). On the other hand, over the Arabian Sea, both forcings contribute to a weakened westerly flow, albeit at slightly different latitudes.

We next analyze the mid- and upper-level circulation associated with the monsoonal flows. The PI 500 hPa vertical velocity field shows a strong ascending flow across the tropics during the monsoon season (Fig. A10b), matching the areas of low SLP shown in Fig. 1d, with the clear exception of the areas under thermal low pressures (e.g., Saudi Arabia and Iran). Subsidence is largely limited to the west Arabian Sea and Somali peninsula (Fig. A10b). Additionally, strong subsidence occurs over the desert regions of the Arabian Peninsula and Iran. Changes in orbital forcing

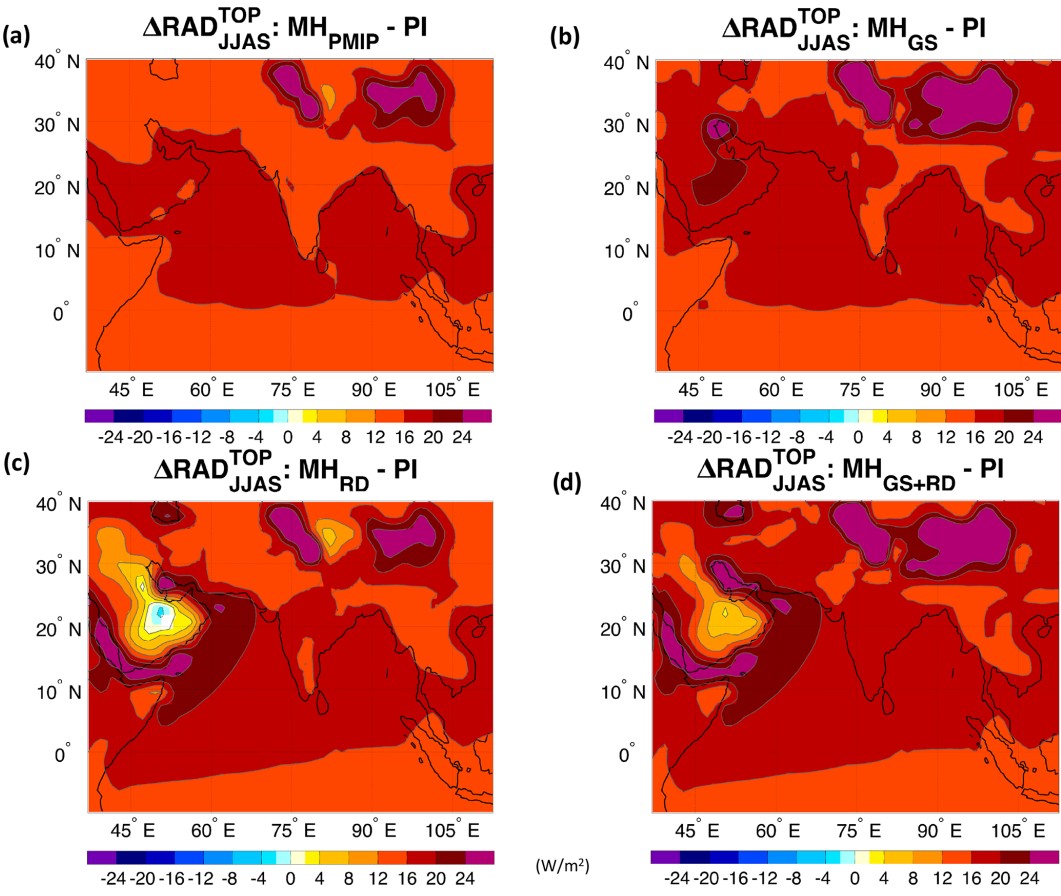

**Figure 5.** Changes in summer (JJAS) top-of-the-atmosphere shortwave radiation ($RAD^{TOP}$; W/m$^2$) for the **(a)** mid-Holocene-only orbital forcing (MH$_{PMIP}$); **(b)** the Sahara greening (MH$_{GS}$); **(c)** the dust-only reduction (MH$_{RD}$); and **(d)** the Sahara greening and dust reduction (MH$_{GS+RD}$) experiments relative to the PI reference simulation. The contour lines follow the color-bar scale (the 0 lines are omitted for clarity). Only differences significant at the 95 % confidence level using the Student $t$ test are shaded.

(MH$_{PMIP}$) drive a strengthened upward motion over the western north-equatorial Indian Ocean, southern India, and the Himalayan foothills (Fig. 9a). This favors cloud formation and is consistent with increased precipitation over these regions (Fig. 2a). Upward anomalies are also found over the climatologically dry southern Arabian Peninsula and part of the Horn of Africa (Fig. 9a). Sahara greening (MH$_{GS}$) intensifies the anomaly pattern seen in the MH$_{PMIP}$ experiment, in particular over northwestern India and the western Indian Ocean, with much stronger increases in upward motions (Fig. A12a). On the other hand, subsidence develops on the lee side of the Western Ghats (Figs. 9b and A12a) due to the stronger easterly anomalies simulated in the MH$_{GS}$ relative to the MH$_{PMIP}$ experiment (Fig. A9a). Reducing Saharan dust emissions (MH$_{RD}$) lead to overall minor and mostly insignificant anomalies over the central SAM region relative to the MH$_{PMIP}$ simulation (cf. Fig. 8a and c, and see also Fig. A12b), except over the southern tip of India where subsidence is increased. However, significant anomalies in the vertical velocity emerge over the Arabian Peninsula relative

to the MH$_{PMIP}$ simulation (Fig. A12b). The result of the Sahara greening and dust reduction forcing (MH$_{GS+RD}$) over Asia is to a great extent a linear combination of the two separate forcings (Fig. 9d), as was indeed the case for the other variables analyzed here.

We next discuss the upper-level velocity potential and divergent winds, which provide a framework to analyze the regional anomalies in the context of the large-scale tropical overturning circulation. The PI experiment shows a divergent flow emanating from southeast Asia towards the surrounding Asian monsoon regions (contour lines in Fig. 10), which is consistent with the low SLP there (Fig. 1d). In the MH$_{PMIP}$, the whole pattern of velocity potential and the centers of divergence/convergence are shifted westward (Fig. 10a), with dipole anomalies centered over northern Africa and the Arabian Peninsula (negative velocity potential/divergence) and South America (positive velocity potential/convergence). The divergence over the northwestern Indian subcontinent is strengthened, which implies an intensified low-level convergence and hence stronger precipitation

https://doi.org/10.5194/cp-17-1-2021

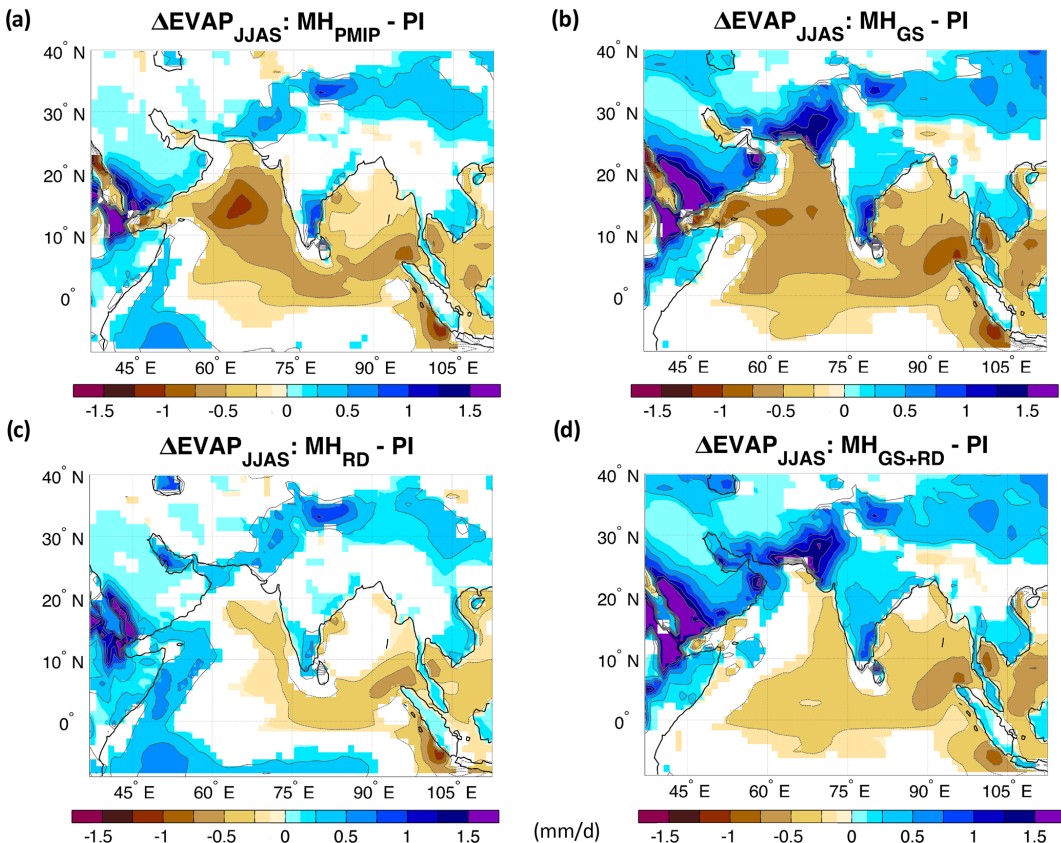

**Figure 6.** Changes in summer (JJAS) evapotranspiration (EVAP; mm/d) for the **(a)** mid-Holocene-only orbital forcing (MH$_{PMIP}$); **(b)** the Sahara greening (MH$_{GS}$); **(c)** the dust-only reduction (MH$_{RD}$); and **(d)** the Sahara greening and dust reduction (MH$_{GS+RD}$) experiments relative to the PI reference simulation. The contour lines follow the color-bar scale (the 0 lines are omitted for clarity). Only differences significant at the 95 % confidence level using the Student $t$ test are shaded.

in the region. The greening of the Sahara (MH$_{GS}$) further intensifies the anomaly pattern seen in the MH$_{PMIP}$ experiment (cf. Fig. 9a and b, and see also Fig. A13a). The dust reduction experiment contributes to a strong positive anomaly in velocity potential over the Arabian Sea relative to MH$_{PMIP}$ (cf. Fig. 9a and c, and see also Fig. A13b), thus weakening upper tropospheric divergence and the lower tropospheric convergence. The Green Sahara reduced dust (MH$_{GS+RD}$) experiment resembles the MH$_{GS}$ forcing, but the anomalies are reduced due to the effect of dust reduction (Figs. 10 and A13c).

The anomalies in velocity potential are negative over both India and the Bay of Bengal, albeit with smaller magnitudes over the latter region. Therefore, the decrease in precipitation over the Bay of Bengal cannot be explained by the changes in upper-level velocity potential/divergence alone. To understand the effect of the greening of the Sahara and the reduction of dust concentrations (MH$_{GS}$, MH$_{RD}$, and MH$_{GS+RD}$) on precipitation over the Bay of Bengal, we consider the rainfall over the western equatorial Indian Ocean (WEIO, 5° S–5° N, 50–65° E) and northeastern Africa (NEA, 10–20° N, 30–45° E). Anomalous convective heating over these regions in response to changes in Earth's precession can drive a

Matsuno–Gill-like response in the low-level winds over the Indian Ocean (Jalihal et al., 2019b). The anomalous easterlies extend into the Bay of Bengal, reducing the wind speed there and leading to a reduction in surface latent heat fluxes. This further leads to a decrease in the net energy flux into the atmosphere (top plus bottom) over the Bay of Bengal. Since precipitation is proportional to the net energy flux into the atmosphere, precipitation over the Bay of Bengal decreases (Jalihal et al., 2019b). The vegetation and dust forcings further modulate the precipitation over the WEIO and NEA, resulting in a corresponding change in precipitation over the Bay of Bengal. MH$_{GS}$ shows the largest increase in precipitation over the WEIO and NEA (Fig. 11a). Proportionately, the decrease in latent heat flux over the Bay of Bengal is also the largest. On the other hand, the weakest increase in precipitation over the WEIO and NEA regions is simulated in the MH$_{RD}$. The associated reduction in latent heat flux over the Bay of Bengal is also the smallest. As the latent heat flux decreases, it leads to a larger reduction in precipitation over the Bay of Bengal (Fig. 11b). This change in latent heat flux is due to the impact of precipitation over the WEIO and NEA on wind speed over the Bay of Bengal (Fig. A14). Our simu-

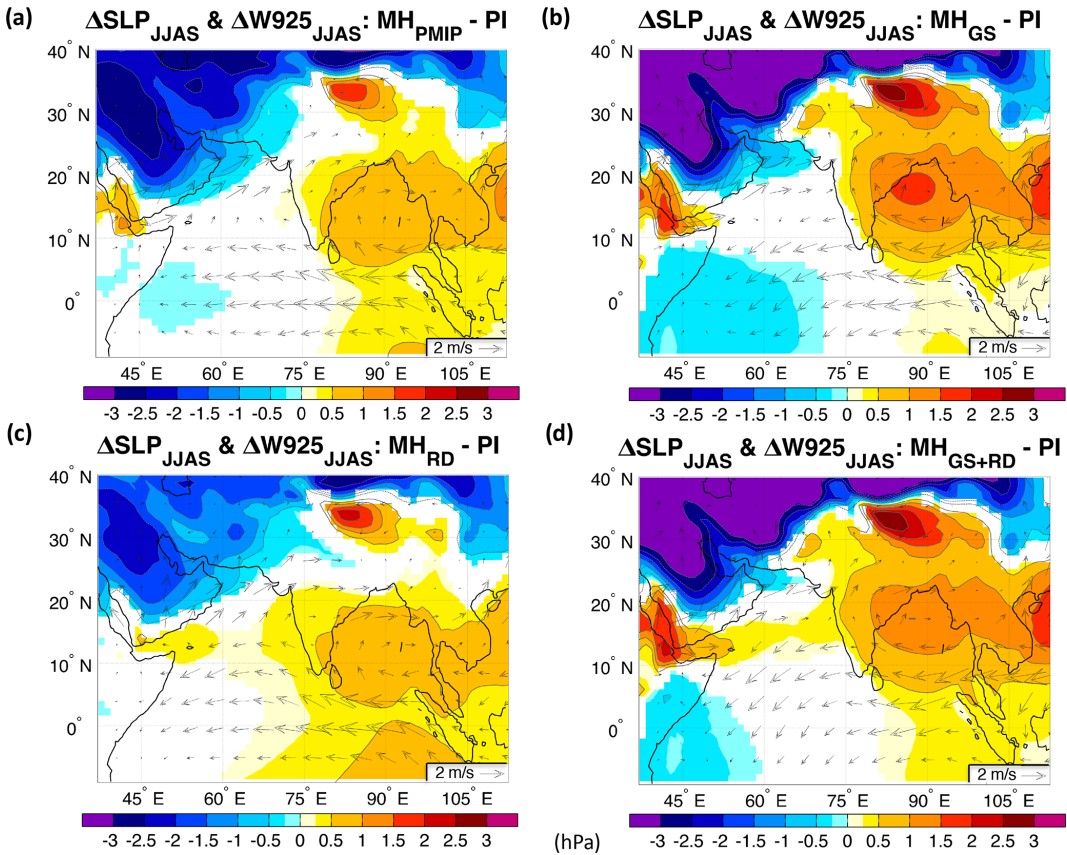

**Figure 7.** Changes in summer (JJAS) sea level pressure (shadings, SLP; hPa) and 925 hPa wind (arrows, W925; m/s) for the **(a)** mid-Holocene-only orbital forcing (MH$_{PMIP}$); **(b)** the Sahara greening (MH$_{GS}$); **(c)** the dust-only reduction (MH$_{RD}$); and **(d)** the Sahara greening and dust reduction (MH$_{GS+RD}$) experiments relative to the PI reference simulation. The contour lines follow the color-bar scale (the 0 lines are omitted for clarity). Only SLP differences significant at the 95 % confidence level using the Student $t$ test are shaded.

lations show a linear relationship between precipitation over the WEIO and NEA, and precipitation over the Bay of Bengal (Fig. 11c).

We conclude our analysis by investigating the changes in the upper-level (200 hPa) jet (Fig. 12). In the PI experiment, the core of the subtropical jet is located over western Asia and the exit of jet is located over northeastern China (contour lines in Fig. 12). In the MH$_{PMIP}$ simulation, the jet is shifted northwards, with an overall weakening to the south and a strengthening confined to the northward side of the exit of the jet streak (Fig. 12a). The Sahara greening (MH$_{GS}$) leads to an accelerated westerly flow at the jet entrance but an overall slowing down at the jet exit together with a further increase in the northward shift relative to the MH$_{PMIP}$ experiment (Fig. 12b). These changes cause a slight tilt in the jet that favors more aloft divergence over northern India and Pakistan as also seen in Fig. 9b, which in turn favors increased rainfall in the region. The dust reduction (MH$_{RD}$) leads to a pattern anomaly very similar to the MH$_{PMIP}$ experiment – albeit weaker (cf. Fig. 12a and c). The effect of the combined forcings (MH$_{GS+RD}$) is dominated by the MH$_{GS}$ pattern (Fig. 12d) and in this case the anomalies are even larger than in the MH$_{GS}$ case. This is likely due to the increase in temperature gradient between low and high latitudes relative to the MH$_{GS}$ case (not shown).

## 3.3 Model–proxy intercomparison

To evaluate the model performance when accounting for Sahara greening and reduction in airborne dust concentrations, we compare our simulations to the available marine and terrestrial paleoclimate archives. We focus on the most apparent dissimilarities between the sensitivity experiments and the standard MH simulation (MH$_{PMIP}$) where only orbital forcing is considered. While our simulations are centered at 6000 BP, they should be seen as indicative of the wet early–middle Holocene rather than a snapshot of exactly 6000 BP, which appears to be a period of transition in particular for Indian terrestrial records (e.g., Prasad et al., 1997).

Notable differences in summer precipitation between the four simulations occur over western India (Fig. 2), which shows substantially wetter conditions in MH$_{GS}$ and MH$_{GS+RD}$, compared to the MH$_{PMIP}$ experiment in that re-

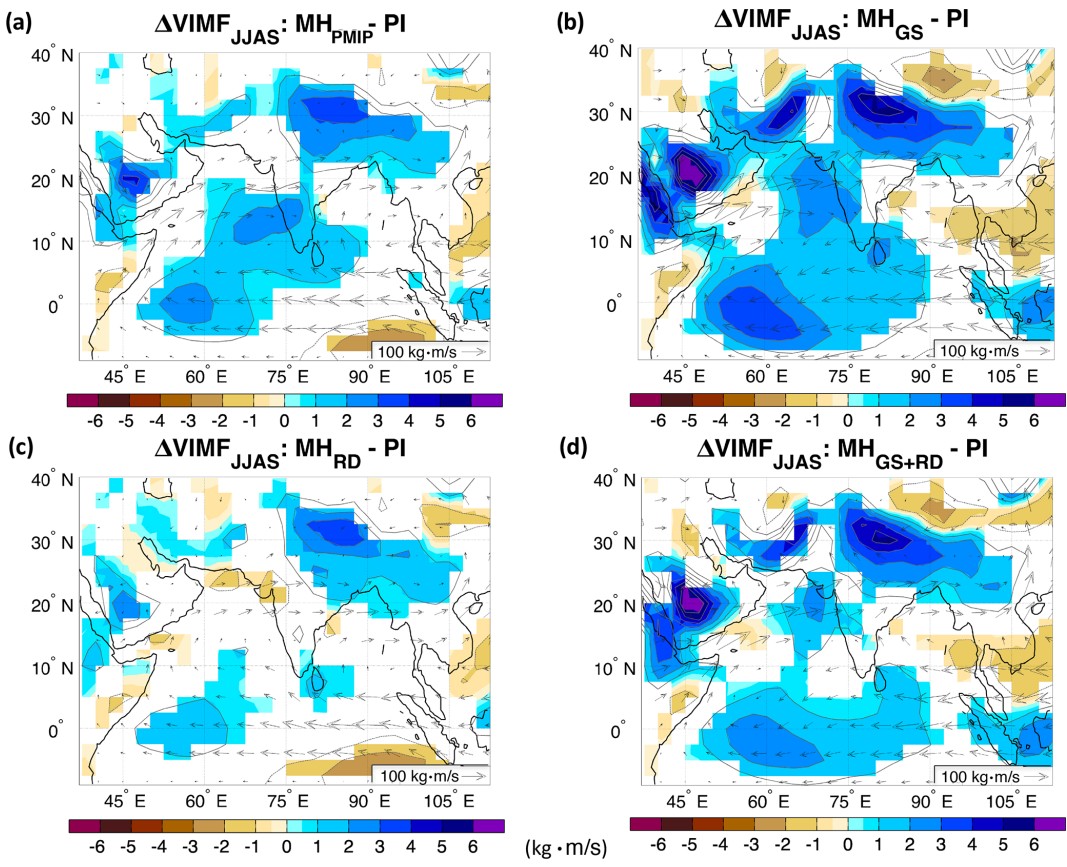

**Figure 8.** Changes in summer (JJAS) vertically integrated (from 1000 to 300 hPa) horizontal moisture flux (VIMF; kg m/s) for the **(a)** mid-Holocene-only orbital forcing (MH$_{PMIP}$); **(b)** the Sahara greening (MH$_{GS}$); **(c)** the dust-only reduction (MH$_{RD}$); and **(d)** the Sahara greening and dust reduction (MH$_{GS+RD}$) experiments relative to the PI reference simulation. The contour lines follow the color-bar scale (the 0 lines are omitted for clarity). The arrows represent the zonal and meridional components of the moisture flux. Only differences significant at the 95 % confidence level using the Student $t$ test are shaded.

gion. Nal Sarovar, a brackish lake bordering the Thar Desert, appears to have been wetter than today around 6200 BP, with a drying tendency towards the end of the MH (Prasad et al., 1997). There is evidence for a substantial pluvial between ∼ 9 and 6 ka TS5 farther north in the core of the Thar (Deotare et al., 2004; Gill et al., 2015; and references therein), and a reduced dimension analysis suggests that reconstructed tropical Pacific SSTs alone could have driven a 60 % increase in precipitation there during the early Holocene (see Fig. 5 in Gill et al., 2017). However, Gill et al. (2017) inferred winds and the precipitation over India using exclusively a proxy-based reconstruction of the tropical Pacific SSTs, assuming modern teleconnections. The MH$_{PMIP}$ experiment simulates a localized rainfall increase in the region of the Thar Desert above 40 %–50 %, whereas the MH$_{GS+RD}$ suggests a more intense and widespread increase in precipitation (Fig. 2) over western and northwestern India, even though the monsoonal flow is weaker compared to MH$_{PMIP}$ (Fig. A4). This suggests that the modern teleconnections may not precisely hold in the past, and the inferred changes based on only tropi-

cal Pacific SST patterns may underestimate the total rainfall changes during the early and middle Holocene over northwestern India.

Another region where our simulations show divergent results is southwestern coastal India. There, the MH$_{GS+RD}$ experiment shows drier conditions relative to PI, while the MH$_{PMIP}$ shows wetter conditions (Fig. 2). Paleoclimate archives from the Nilgiri Mountains, in the Western Ghats at the eastern edge of the simulated dry anomaly, suggest that the region was wetter between 12 000 and 10 000 BP and then gradually became drier during the mid-Holocene relative to today (Sukumar et al., 1993). Hence, accounting for the greening of the Sahara may improve the precipitation anomaly pattern seen during the mid-Holocene over India; however, a systematic model validation is not currently possible due to the paucity of available paleoclimate archives and their large uncertainties.

With respect to changes in SSTs, the MH$_{GS+RD}$ experiment simulates a warming of the Arabian Sea and Bay of Bengal summer SSTs relative to PI, while little change

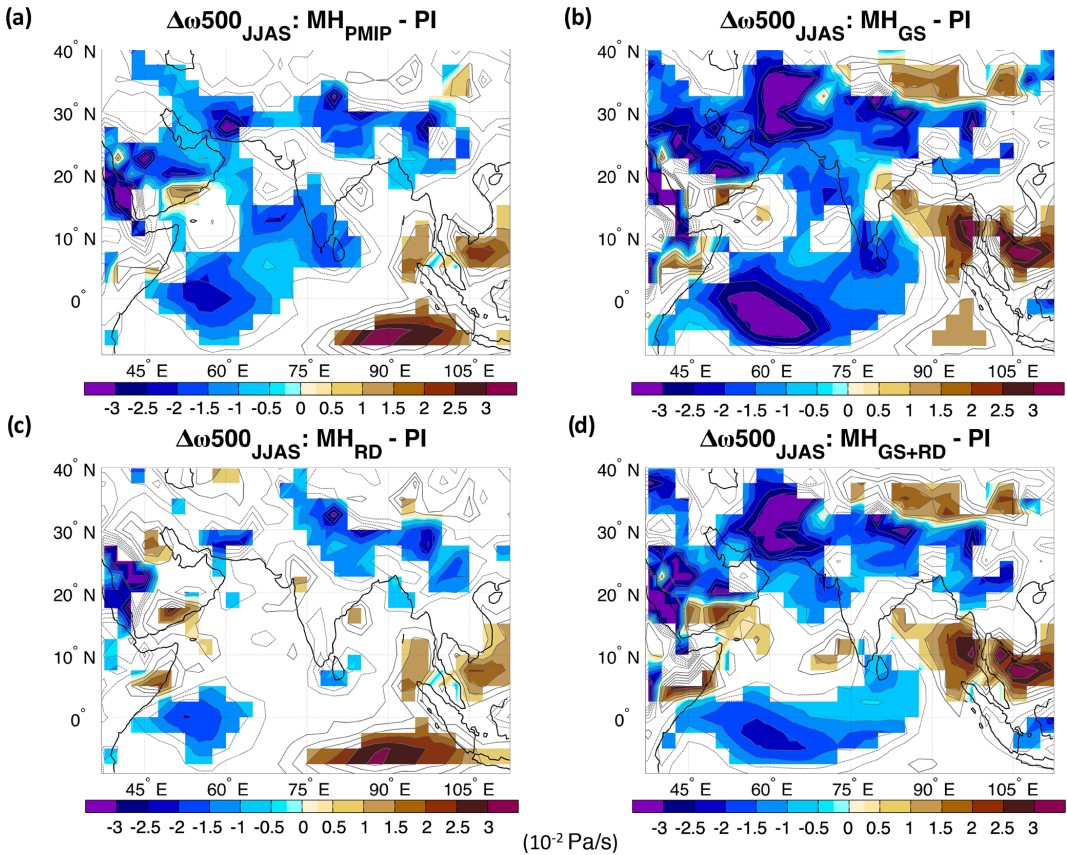

**Figure 9.** Changes in summer (JJAS) vertical pressure velocity at 500 hPa ($\omega 500$; Pa/s) for the **(a)** mid-Holocene-only orbital forcing (MH$_{PMIP}$); **(b)** the Sahara greening (MH$_{GS}$); **(c)** the dust-only reduction (MH$_{RD}$); and **(d)** the Sahara greening and dust reduction (MH$_{GS+RD}$) experiments relative to the PI reference simulation. The contour lines follow the color-bar scale (the 0 lines are omitted for clarity). Only differences significant at the 95 % confidence level using the Student $t$ test are shaded.

or a slight cooling is simulated in the MH$_{PMIP}$ experiment (Fig. 4). In the Arabian Sea, proxy evidence for widespread mid-Holocene SST warming is lacking. Dahl and Oppo (2006) showed that the early Holocene (at around 8000 BP) was $1.4 \pm 1.3$ °C cooler than the late Holocene, on the basis of Mg/Ca in the planktic foraminifer *Globigerinoides ruber* from 12 cores spanning much of the basin. Their only core showing a slight warming at 8000 BP ($+1.0$ °C) was situated off the Horn of Africa, a region with robust warming in all four mid-Holocene simulations. Other *G. ruber* and *Trilobatus sacculifer* Mg/Ca records corroborate modest cooling ($\sim 0$ to $-1$ °C) during the mid-Holocene (around 6000 BP) in the eastern Arabian Sea (Anand et al., 2008; Banakar et al., 2010; Govil and Naidu, 2010), with slight warming off the coast of southwest India (Saraswat et al., 2013) and again off the Horn of Africa (Anand et al., 2008). Alkenones document 0 to 1 °C cooling in the northern Arabian Sea during this time period (Böll et al., 2015; Schulte and Müller, 2001), with negligible change off the Arabian Peninsula (Huguet et al., 2006; Rostek et al., 1997) and southwest India (Sonzogni et al., 1998). Regional

Mg/Ca and alkenone compilations by Gaye et al. (2018) suggest that no sector of the Arabian Sea was warmer at 6 ka [TS6], with the possible exception of south of India, which also warms slightly in all four simulations. Alkenones from the northern Bay of Bengal (Lauterbach et al., 2020) and *G. ruber* Mg/Ca from the southern Bay of Bengal (Raza et al., 2017) indicate <1 °C cooling during the mid-Holocene. Records from the Andaman Sea in the northeastern Indian Ocean show contrasting results, with some presenting a slight cooling (*G. ruber*; Rashid et al., 2007) and others a slight warming (*T. sacculifer*; Gebregiorgis et al., 2016), conceivably due to habitat differences between the planktic foraminifera species used. Overall, given that both simulated and proxy-documented changes are mostly <1 °C in the Arabian Sea and Bay of Bengal, it is difficult to draw definitive conclusions on the accuracy of the model simulations there.

South of the Equator, west of Sumatra, the MH$_{PMIP}$ simulation produces a stronger cooling (>1 °C) that disappears in the MH$_{GS+RD}$ experiment (Fig. 4). Here, alkenones are more consistent with MH$_{PMIP}$, albeit with a more modest cooling of 0.5 to 1 °C (Li et al., 2016; Lückge et al., 2009).

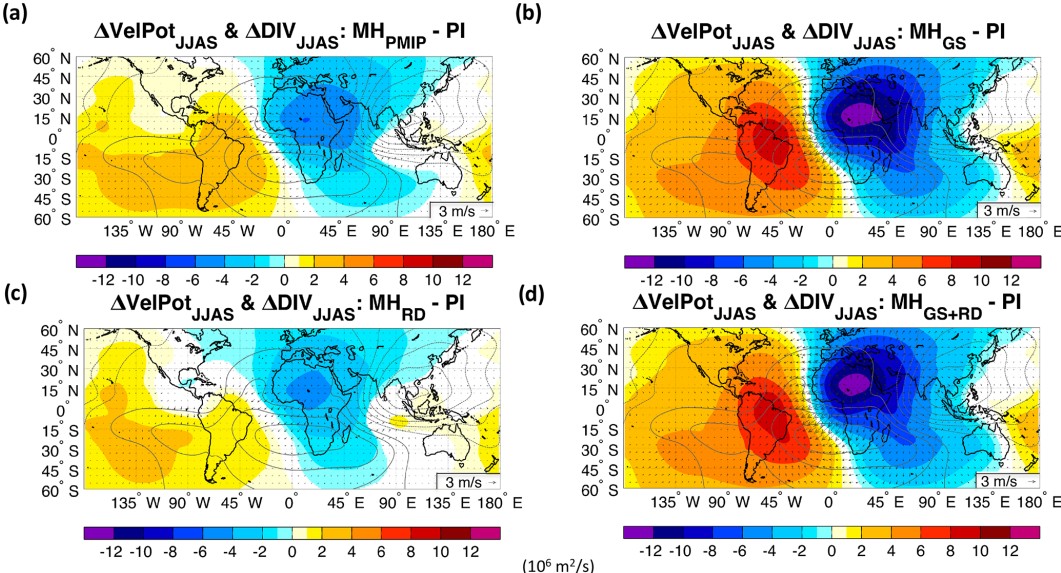

**Figure 10.** Changes in summer (JJAS) velocity potential (VelPot – shadings; $10^6$ m$^2$/s TS4) and divergence wind (DIV – arrows; m/s) at 200 hPa for the **(a)** mid-Holocene-only orbital forcing (MH$_{PMIP}$); **(b)** the Sahara greening (MH$_{GS}$); **(c)** the dust-only reduction (MH$_{RD}$); and **(d)** the Sahara greening and dust reduction (MH$_{GS+RD}$) experiments relative to the PI reference simulation. The contour lines show the climatological summer velocity potential of the PI experiment. Only differences significant at the 95 % confidence level using the Student $t$ test are shaded.

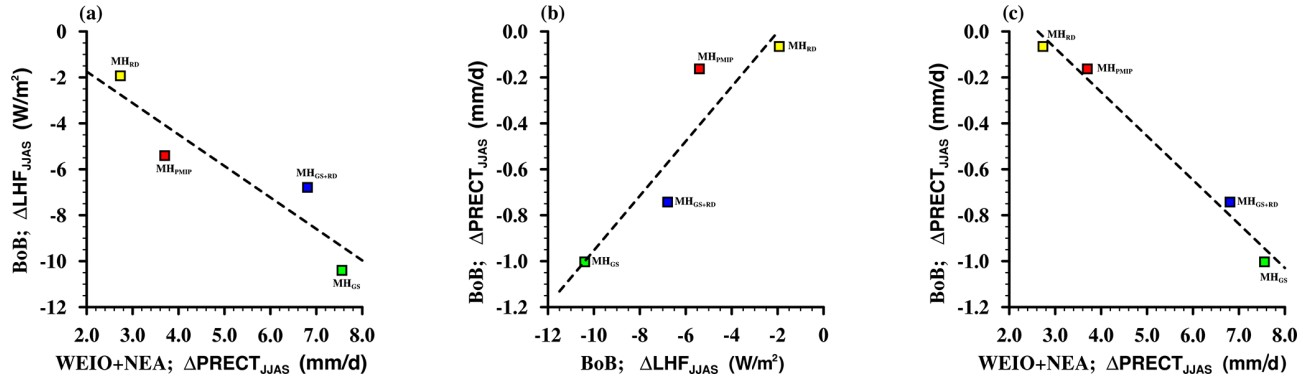

**Figure 11.** Scatter plot of summer (JJAS) changes between **(a)** latent heat flux over the Bay of Bengal (BoB, 10–20° N, 85–95° E; W/m$^2$) and precipitation over the western equatorial Indian Ocean (WEIO, 5° S to 5° N, 50–65° E; mm/d), and northeastern Africa (NEA, 10–20° N, 30–45° E), between **(b)** precipitation and latent heat flux over the BoB, and between **(c)** precipitation over the BoB and over WEIO + NEA. Changes are shown for the mid-Holocene-only orbital forcing (MH$_{PMIP}$) in red, the Sahara greening (MH$_{GS}$) in green, dust-only reduction (MH$_{RD}$) in yellow, and the Sahara greening with dust reduction (MH$_{GS+RD}$) in blue with respect to the PI reference simulation.

However, *G. ruber* Mg/Ca indicates negligible change, more consistent with MH$_{GS+RD}$ (Mohtadi et al., 2010). Seasonal differences in proxy carrier production may explain such differences, with Mg/Ca perhaps being more appropriate for comparison to JJAS simulations, as suggested for the equatorial Pacific (Gill et al., 2016; Timmermann et al., 2014). Finally, south of the Equator, on the western side of the Indian basin off the coast of Tanzania, Mg/Ca reconstruction suggests that SSTs were about 1 to 1.5 °C warmer during the mid-Holocene compared to late Holocene (Kuhnert et al., 2014), which is more consistent with the MH$_{GS+RD}$ experi-

ment (Fig. 4d). This record also shows a rapid SST cooling concomitant with an abrupt retreat of the SAM as suggested by a recently published paleoclimate archives from western Yunnan Plateau in southwestern China (Wang et al., 2020) and northern Laos (Griffiths et al., 2020). Such changes are also synchronous with the end of the African Humid Period (e.g., deMenocal et al., 2000); hence, our simulations suggest that the changes in vegetation over the Sahara and in airborne dust emissions may have played a key role in shaping the evolution of the SAM.

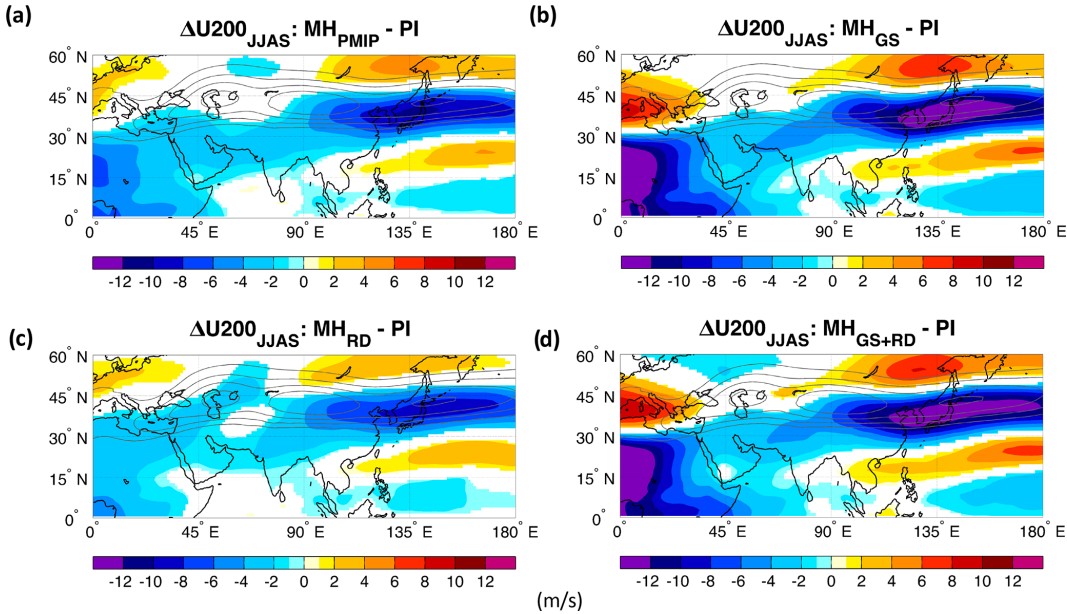

**Figure 12.** Changes in summer (JJAS) zonal wind at 200 hPa (U200; m/s) for the **(a)** mid-Holocene-only orbital forcing ($MH_{PMIP}$); **(b)** the Sahara greening ($MH_{GS}$); **(c)** the dust-only reduction ($MH_{RD}$); and **(d)** the Sahara greening and dust reduction ($MH_{GS+RD}$) experiments relative to the PI reference simulation. The contour lines show the climatological summer zonal wind at 200 hPa for the PI experiment. Only differences significant at the 95 % confidence level using the Student $t$ test are shaded.

## 4 Discussion and conclusions

The mid-Holocene was characterized by a strengthening of the Northern Hemisphere monsoon system (e.g., Sun et al., 2019) due to increased boreal summer insolation. The consequent increase in rainfall led to a greening of several semi-arid and arid regions in northern Africa and Asia (e.g., Campo et al., 1982; Dallmeyer et al., 2013; Fleitmann et al., 2003; Lézine et al., 2011; Tierney et al., 2017) and to a marked reduction in airborne dust emissions (deMenocal et al., 2000; McGee et al., 2013). The largest dust emission decreases are thought to have occurred in northern Africa, where large tracts of what is today the Sahara were vegetated. Understanding this complex set of interrelated changes can provide insights into the mechanisms of monsoonal variability and contribute to strengthening our physical understanding of monsoonal changes in climate projections. However, many modeling efforts for the mid-Holocene have focused only on the impact of solar insolation changes, as this has been the common protocol for climate simulations of this period (Otto-Bliesner et al., 2017; Taylor et al., 2009, 2012), neglecting the feedbacks induced by the altered vegetation, soil properties, and associated dust emissions.

Indeed, the role of reduced dust emissions during the mid-Holocene on local and global climate has only recently been addressed (Hopcroft and Valdes, 2019; Pausata et al., 2016, 2017a, b; Piao et al., 2020; Sun et al., 2019; Thompson et al., 2019), and it has been shown that airborne dust may play an important role in modulating the intensity and ge-

ographical extent of the West African Monsoon (Pausata et al., 2016; Thompson et al., 2019) as well as impacting climate far afield. However, the role of Saharan dust changes in affecting the SAM system has not hitherto been investigated. The key goal of the present study is to fill this knowledge gap by outlining the remote response of the mid-Holocene SAM system to the Sahara greening and associated reduction in airborne dust concentrations.

We analyze a set of simulations where the land cover is changed from desert to shrubland over a large part of northern Africa and dust concentration over the region is reduced by up to 80 % compared to the pre-industrial period (Gaetani et al., 2017; Pausata et al., 2016). We find that a vegetated Sahara – albeit weakening the low-level southwesterly winds – enhances the moisture flux from the Arabian Sea to the northern Indian subcontinent and increases the precipitation in this region compared to a simulation in which only the orbital forcing is considered (Figs. 2, A9, and A11). Reduced dust emissions from the Sahara partially counter the vegetation effect by weakening the thermal low over the Arabian Peninsula and the climatological southwesterlies and subsidence (Figs. 7, 9, A9, A11, and A12). This results in decreased precipitation over India in the mid-Holocene experiment with both changes in vegetation and dust concentration ($MH_{GS+RD}$) compared to the vegetated-Sahara-only case ($MH_{GS}$), especially in the central-southern and western seaboard regions (Figs. 2 and A3). Overall, the SAM rainfall in the $MH_{GS+RD}$ is significantly increased compared to the PI climate as well as to the orbital-forcing-only simulation

(MH$_{PMIP}$). The monsoon season is also extended by several months, particularly in the withdrawal phase (Fig. 3).

Sun et al. (2019) showed that the greening of the Sahara and a reduction in dust emissions significantly influence the Northern Hemisphere land monsoon precipitation, but the largest impact is on the WAM. Here, we show that the SAM is significantly affected by both vegetation changes in northern Africa and dust reduction, and the remote response is about half of the rainfall change simulated locally over northern Africa (cf. Fig. 2 here with Fig. 2 in Pausata et al., 2016). However, the simulated impact of dust changes needs further investigation, as rainfall in tropical regions is strongly affected by the specific prescription of dust optical properties. In particular, the choice of the single scattering albedo $\omega_0$ can significantly alter the effect of dust on precipitation via the so-called "heat pump" effect (Lau et al., 2009). The atmospheric dust layer in which the dust particles are moderate-to-highly absorbing (single scattering albedo $\omega_0 < 0.95$) warms the atmosphere, enhancing deep convection and intensifying the monsoonal precipitation (Lau et al., 2009). In particular, EC-Earth has a single scattering albedo of 0.89 at 550 nm (Table A1). Such a value is too absorbing compared to observations (see Fig. 1 in Albani et al., 2014), and consequently the radiative impact of dust is likely overestimated as also pointed out in Hopcroft and Valdes (2019). Furthermore, Albani and Mahowald (2019) showed how different choices in terms of dust optical properties and size distributions may yield opposite results in terms of rainfall changes. For example, Shi et al. (2019) showed that the dust radiative effect intensifies the SAM, which is opposite to our results. This difference results in a warming of the Tibetan Plateau and central Asia when reducing dust under a Green Sahara, likely associated with a decrease in precipitation in the region.

A further caveat of our work with respect to dust is that we rely on an idealized dust reduction pattern, as opposed to more realistic global dust modulation patterns (e.g., Albani et al., 2015). However, in the EC-Earth simulations, most of the changes in the WAM intensity and the teleconnection to the SAM were associated with changes in surface albedo due to greening of the Sahara. The surface albedo changes were then further enhanced by dust reduction. This rainfall response in the WAM is opposite to what one would expect from a reduced "heat pump" effect (decreased rainfall), suggesting that the "heat pump" effect is overwhelmed by the changes in surface albedo under Green Sahara conditions in EC-Earth simulations. Moreover, previous work (Gaetani et al., 2017) has shown that the results from different dust distributions are very similar and do not alter the qualitative conclusion that dust changes amplify the effects associated with land-surface changes.

A comparison of our simulations with paleoclimate archives points to potential improvements in simulating rainfall over India when including the greening of the Sahara and dust reduction relative to the orbital-forcing-only simulation. In particular, our simulations suggest that the vegetation and dust emission changes may have played an important role in affecting the Indian Ocean temperature and shaping the evolution of the SAM during the termination of the African Humid Period. However, no robust conclusions can be drawn in this respect due to the relative paucity of geographically and temporally referenced, quantitative paleo-precipitation data in the region. A similar difficulty is encountered in evaluating the modeled SST changes. Only some paleo-archives point to closer agreement with the MH$_{GS+RD}$ simulation; however, in general, the amplitudes of SST changes are small relative to proxy uncertainties, making it difficult to provide a systematic model validation.

Finally, in our experiments, we only consider changes in vegetation over northern Africa and its remote impact on SAM. However, proxy archives from the mid-Holocene point to widespread vegetation changes across the globe, with expanded forest cover in Eurasia (Prentice et al., 1998; Tarasov et al., 1998) and greener southern and eastern Asia (Dykoski et al., 2005; Fleitmann et al., 2003; Thompson et al., 1997; Zhang et al., 2014). Swann et al. (2012, 2014) show that in their model the remote forcing from expanded forest cover in Eurasia during the mid-Holocene shifts the intertropical convergence zone northward, resulting in an enhancement of precipitation over northern Africa that is greater than that resulting from orbital forcing and local vegetation alone. Using idealized deforestation experiments in the tropics and temperate regions, Devaraju et al. (2015) showed that the monsoonal precipitation changes can be more sensitive to remote than local changes in vegetation. Hence, it is possible that the rainfall changes seen in our study may be further modulated by vegetation changes in Europe and Asia. Therefore, it is critical that the Earth system modeling community conducts a concerted effort to include reconstructed vegetation distributions and dust concentrations when simulating the mid-Holocene climate.

## Appendix A: Model validation

In order to evaluate the performance of the EC-Earth model used here in reproducing the SAM dynamics, we compare our PI simulation to surface temperature and precipitation data from ECMWF's ERA5 reanalysis product (Hersbach et al., 2020) and gridded observational products. Long-term precipitation rates from ERA5 compare favorably with NASA's Tropical Rainfall Measuring Mission (TRMM) multi-satellite precipitation analysis (Hersbach et al., 2020; Huffman et al., 2010), and over the Indian subcontinent differences between ERA5 and the Global Precipitation Climatology Project (GPCP) gridded observational dataset (Adler et al., 2018) are mostly below 0.5 mm/d (Figs. A1 and A2). Good agreement is also found between ERA5 temperatures and the Climatic Research Unit (CRU) dataset (Harris et al., 2020) and ERA5 improves in this respect over previous datasets (Hersbach et al., 2020).

EC-Earth's PI simulation in general underestimates rainfall over the northeastern Indian subcontinent and overestimates it over the western side. The model further presents a large cold bias (Fig. A1).

**Table A1.** Aerosol optical depth, single scattering albedo, and composition of the mineral dust for a relative humidity of 50 %.

| Type | RH (%) | AOD at 550 nm | SSA ($\omega_0$) | Component | Number (cm$^{-3}$) | Mass (µg/m$^3$) |
|---|---|---|---|---|---|---|
| "Desert" dust-like | 50 | 0.037 | 0.888 | Total | 2300 | 225.8 |
| | | | | Water soluble | 2000 | 4.0 |
| | | | | Mineral (nuclei) | 269.5 | 7.5 |
| | | | | Mineral (accum.) | 30.5 | 168.7 |
| | | | | Mineral (coarse) | 0.142 | 45.6 |

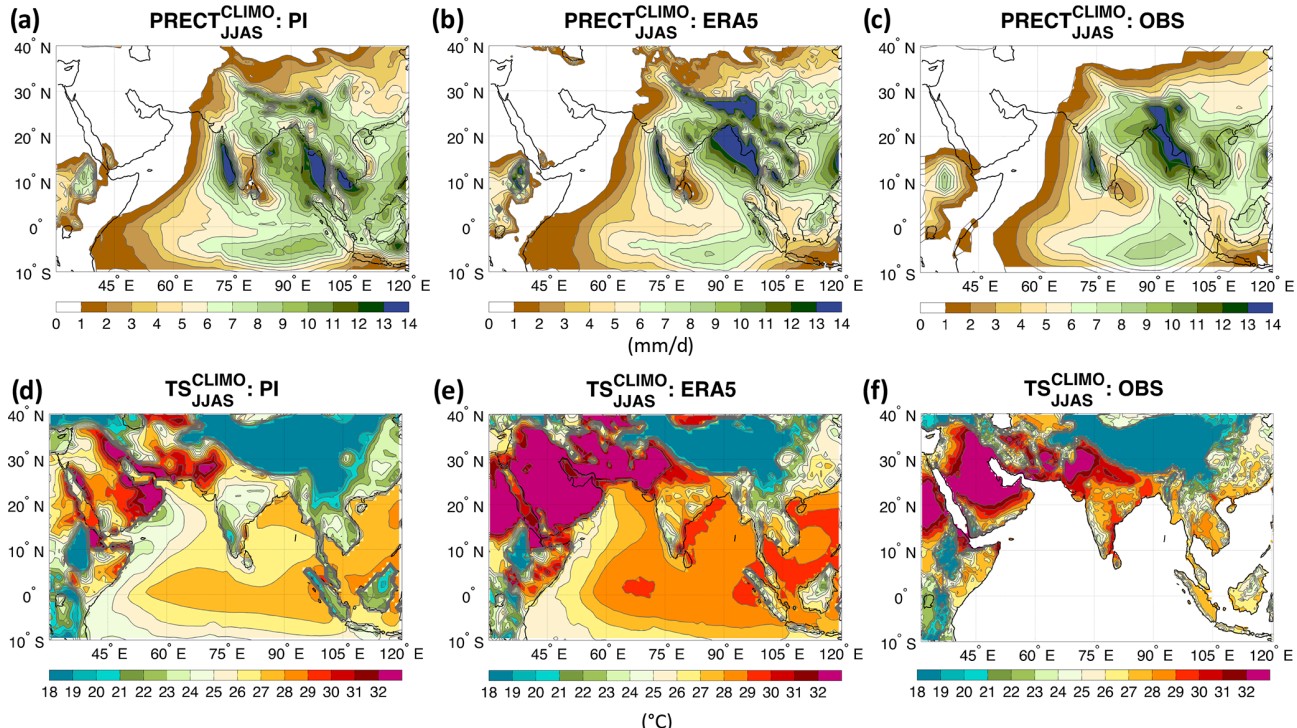

**Figure A1.** Climatological summer (JJAS) precipitation (PRECT; mm/d) from **(a)** the PI simulation, **(b)** the ERA5 reanalysis for the period 1979–2018, and **(c)** the Global Precipitation Climatology Project (GPCP) version 2.3 for the period 1979–2018. Climatological summer (JJAS) surface temperature (TS; °C) from **(d)** the PI and **(e)** the ERA5 reanalysis for the period 1979–2018; and **(f)** near-surface temperature from the Climatic Research Unit (CRU) time series version 4.04 for the period 1979–2018.

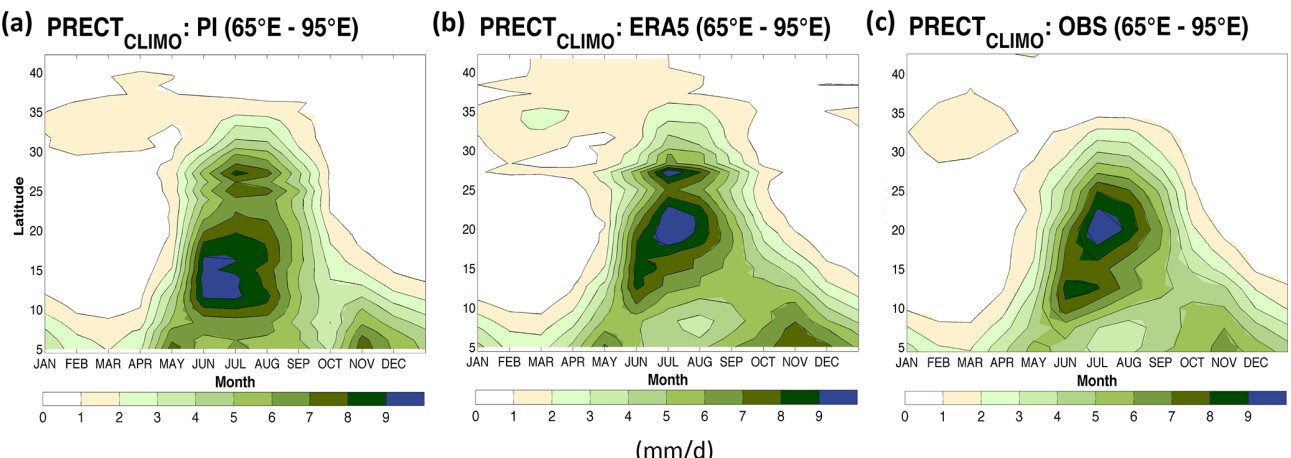

**Figure A2.** Climatological seasonal cycle of zonal-mean precipitation (PRECT; mm/d) between 65 and 95° E from **(a)** the PI simulation, **(b)** the ERA5 reanalysis for the period 1979–2018, and **(c)** the Global Precipitation Climatology Project (GPCP) version 2.3 for the period 1979–2018.

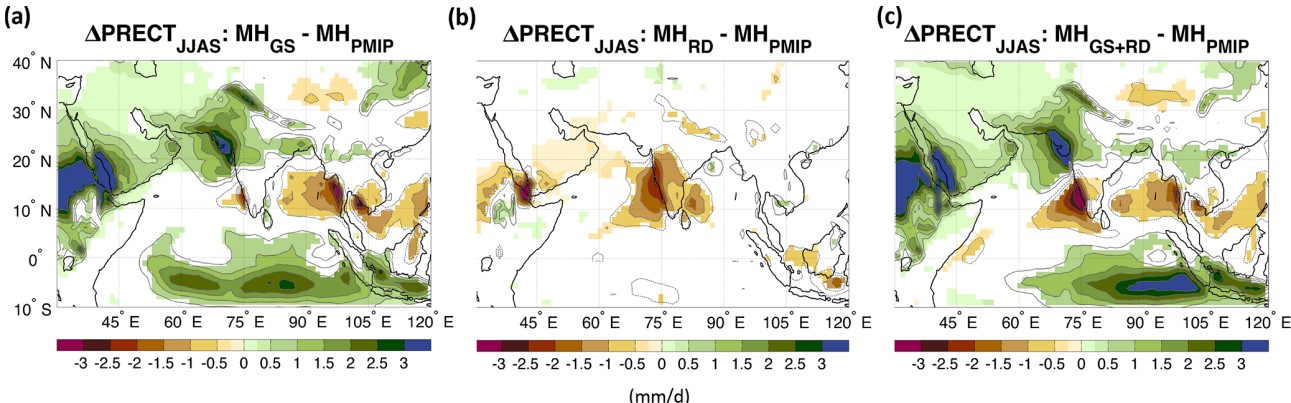

**Figure A3.** Changes in summer (JJAS) precipitation (PRECT; mm/d) for **(a)** the mid-Holocene Green Sahara (MH$_{GS}$); **(b)** the dust-only reduction (MH$_{RD}$); and **(c)** the Sahara greening and dust reduction (MH$_{GS+RD}$) experiments relative to the mid-Holocene-only orbital forcing (MH$_{PMIP}$) simulation. The contour lines follow the color-bar scale (the 0 lines are omitted for clarity). Only differences significant at the 95 % confidence level using the Student $t$ test are shaded.

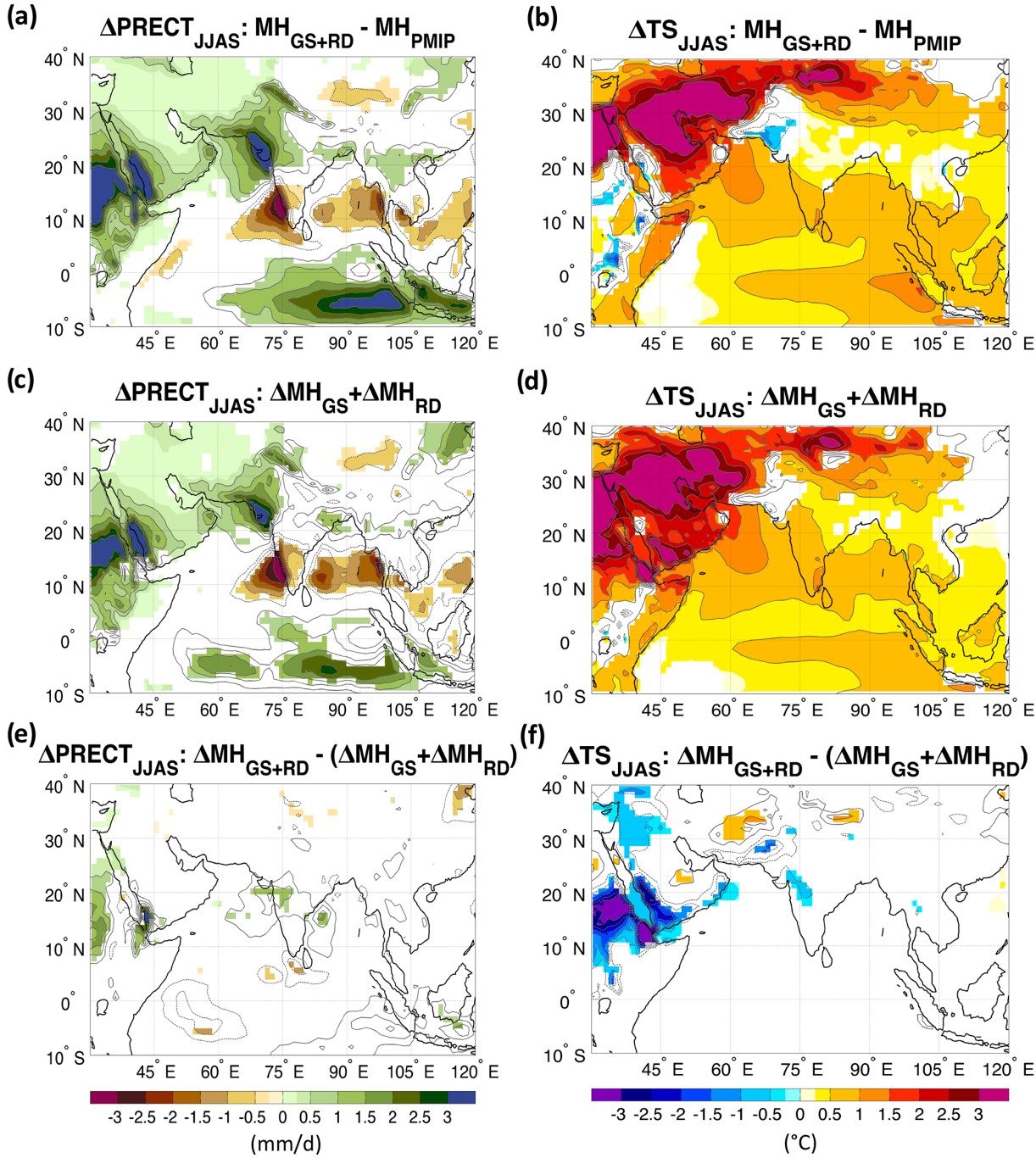

**Figure A4. (a)** Summer (JJAS) precipitation (PRECT, mm/d) and **(b)** surface temperature (TS; °C) anomalies between the $MH_{GS+RD}$ and the $MH_{PMIP}$ experiments. **(c)** The sum of $MH_{GS}$ and $MH_{RD}$ precipitation and **(d)** surface temperature anomalies relative to the reference $MH_{PMIP}$ experiment. **(e)–(f)** Difference between panels **(a)** and **(c)**, and **(b)** and **(d)**, respectively. Only differences significant at the 95 % confidence level using the Student $t$ test are shaded.

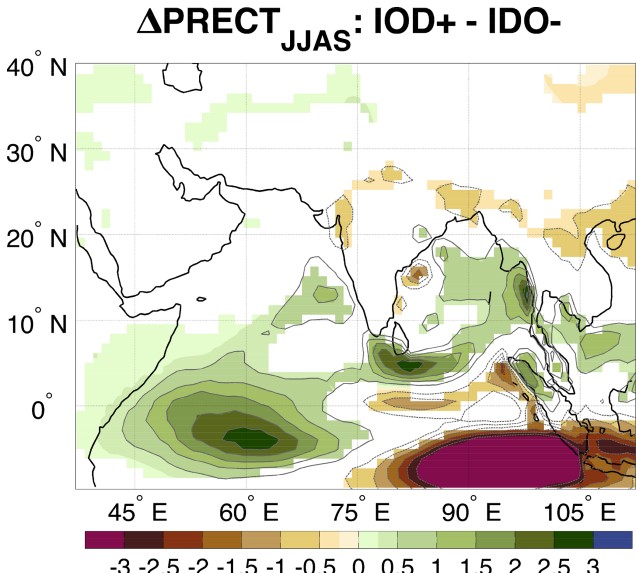

**Figure A5.** Difference between the positive and negative phases of the Indian Ocean dipole (IOD) composite for summer (JJAS) precipitation (PRECT, mm/d) in the PI experiment. Only differences significant at the 95 % confidence level using the Student $t$ test are shaded. The contours follow the color-bar intervals (solid for positive and dashed for negative anomalies; the zero line is omitted).

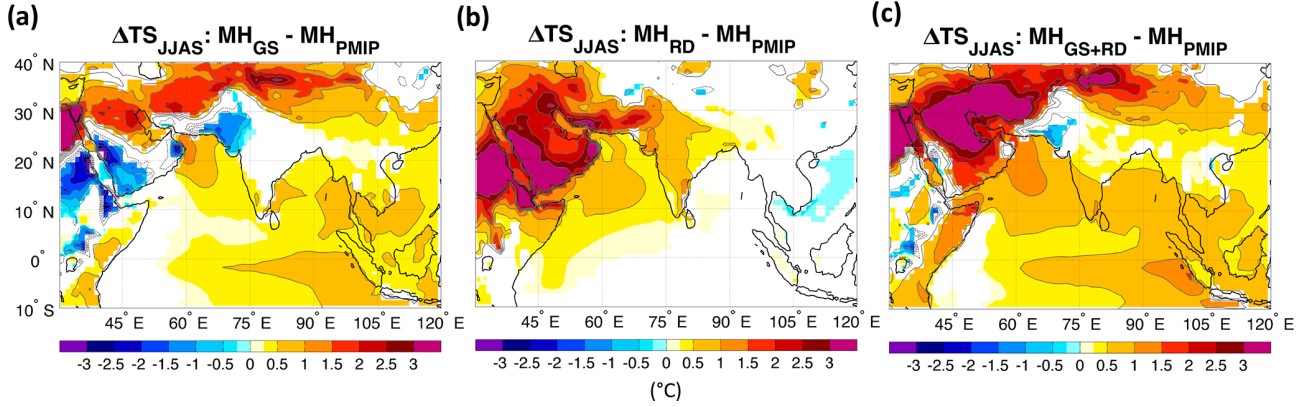

**Figure A6.** Changes in summer (JJAS) surface temperature (TS; °C) for **(a)** the mid-Holocene Green Sahara (MH$_{GS}$), **(b)** the dust-only reduction (MH$_{RD}$); and **(c)** the Sahara greening and dust reduction (MH$_{GS+RD}$) experiments relative to the mid-Holocene-only orbital forcing (MH$_{PMIP}$) simulation. The contour lines follow the color-bar scale (the 0 lines are omitted for clarity). Only differences significant at the 95 % confidence level using the Student $t$ test are shaded.

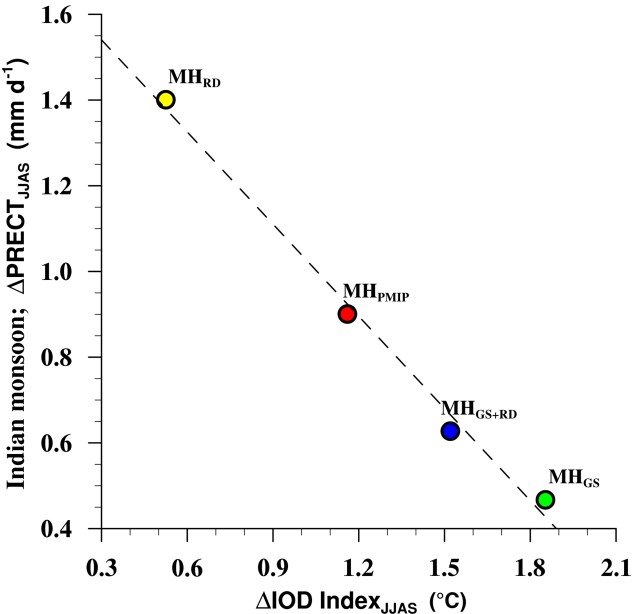

**Figure A7.** Scatter plot of summer (JJAS) changes between precipitation over India (10–30° N, 70–85° E; land only, mm/d) and the IOD index (K). The IOD index is the difference between the western equatorial Indian Ocean (10° S–10° N, 50–70° E) and eastern equatorial Indian Ocean (10° S–0° N, 90–110° E) sea surface temperature anomaly. The changes are shown for the mid-Holocene-only orbital forcing (MH$_{PMIP}$) in red, the Sahara greening (MH$_{GS}$) in green, dust-only reduction (MH$_{RD}$) in yellow, and the Sahara greening with dust reduction (MH$_{GS+RD}$) in blue with respect to the PI reference simulation.

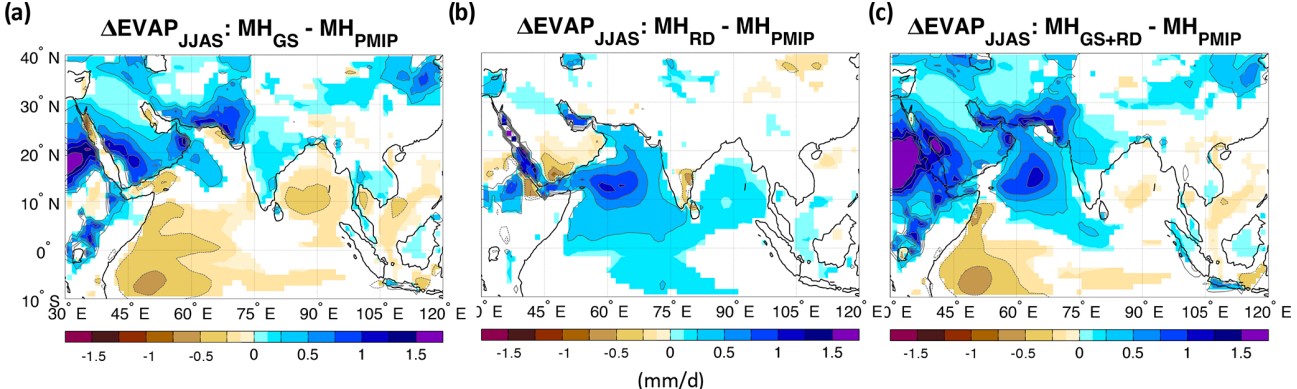

**Figure A8.** Changes in summer (JJAS) evapotranspiration (EVAP; mm/d) for **(a)** the mid-Holocene Green Sahara (MH$_{GS}$); **(b)** the dust-only reduction (MH$_{RD}$); and **(c)** the Sahara greening and dust reduction (MH$_{GS+RD}$) experiments relative to the mid-Holocene-only orbital forcing (MH$_{PMIP}$) simulation. The contour lines follow the color-bar scale (the 0 lines are omitted for clarity). Only differences significant at the 95 % confidence level using the Student $t$ test are shaded.

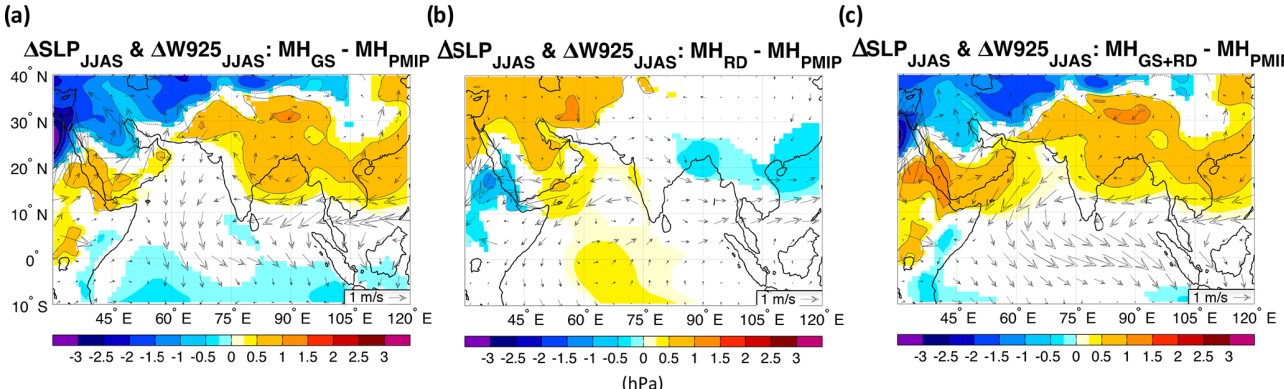

**Figure A9.** Changes in summer (JJAS) sea level pressure (shadings, SLP; hPa) and 925 hPa wind (arrows, W925; m/s) for **(a)** the mid-Holocene Green Sahara ($MH_{GS}$); **(b)** the dust-only reduction ($MH_{RD}$); and **(c)** the Sahara greening and dust reduction ($MH_{GS+RD}$) experiments relative to the mid-Holocene-only orbital forcing ($MH_{PMIP}$) simulation. The contour lines follow the color-bar scale (the 0 lines are omitted for clarity). Only SLP differences significant at the 95 % confidence level using the Student $t$ test are shaded.

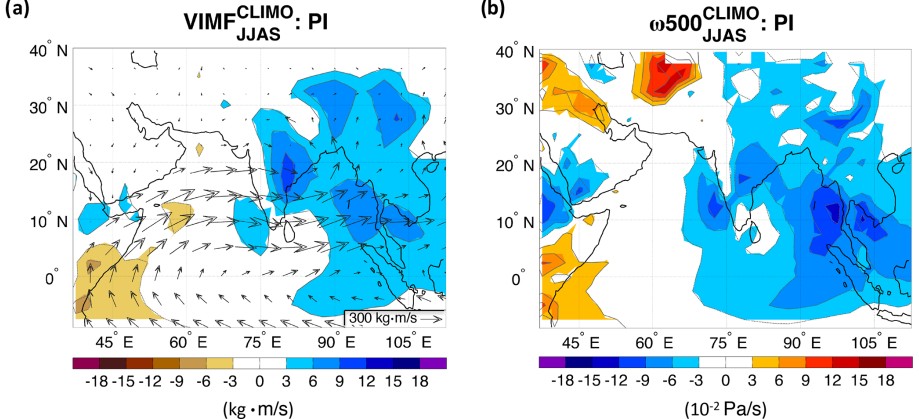

**Figure A10.** Climatological summer (JJAS) **(a)** vertical integrated horizontal moisture flux (VIMF, kg m/s) with the arrows representing the zonal and meridional components of the moisture flux; **(b)** vertical pressure velocity at 500 hPa ($\omega$500, $10^{-2}$ Pa/s TS7), in the PI simulation.

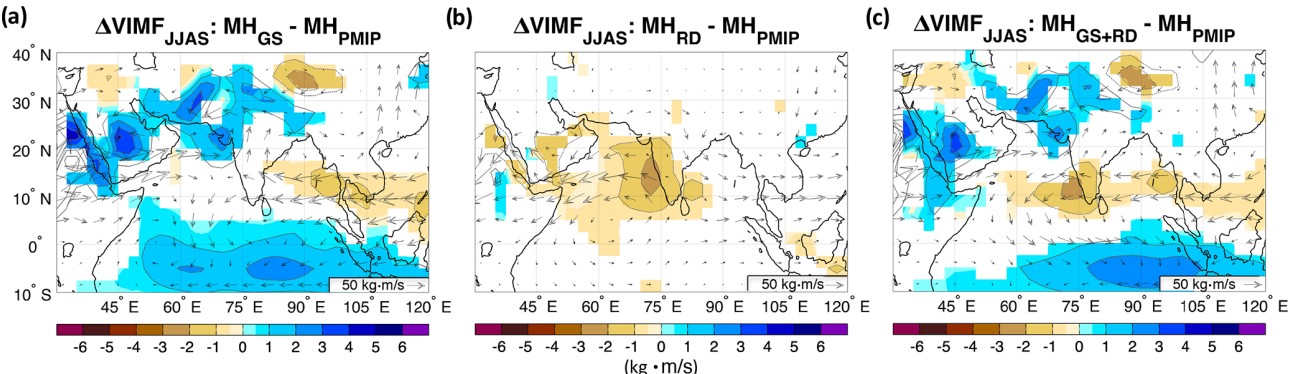

**Figure A11.** Changes in summer (JJAS) vertically integrated (from 1000 to 300 hPa) horizontal moisture flux (VIMF; kg m/s) for **(a)** the mid-Holocene Green Sahara ($MH_{GS}$); **(b)** the dust-only reduction ($MH_{RD}$); and **(c)** the Sahara greening and dust reduction ($MH_{GS+RD}$) experiments relative to the mid-Holocene-only orbital forcing ($MH_{PMIP}$) simulation. The contour lines follow the color-bar scale (the 0 lines are omitted for clarity). The arrows represent the zonal and meridional components of the moisture flux. Only differences significant at the 95 % confidence level using the Student $t$ test are shaded.

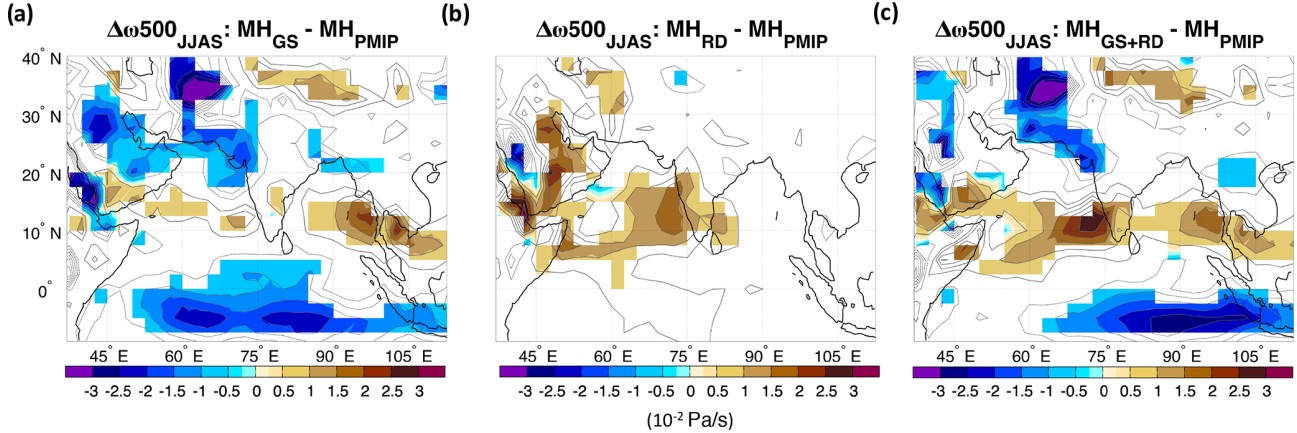

**Figure A12.** Changes in summer (JJAS) vertical pressure velocity at 500 hPa ($\omega$500; Pa/s TS8) for **(a)** the mid-Holocene Green Sahara ($MH_{GS}$); **(b)** the dust-only reduction ($MH_{RD}$); and **(c)** the Sahara greening and dust reduction ($MH_{GS+RD}$) experiments relative to the mid-Holocene-only orbital forcing ($MH_{PMIP}$) simulation. The contour lines follow the color-bar scale (the 0 lines are omitted for clarity). Only differences significant at the 95 % confidence level using the Student $t$ test are shaded.

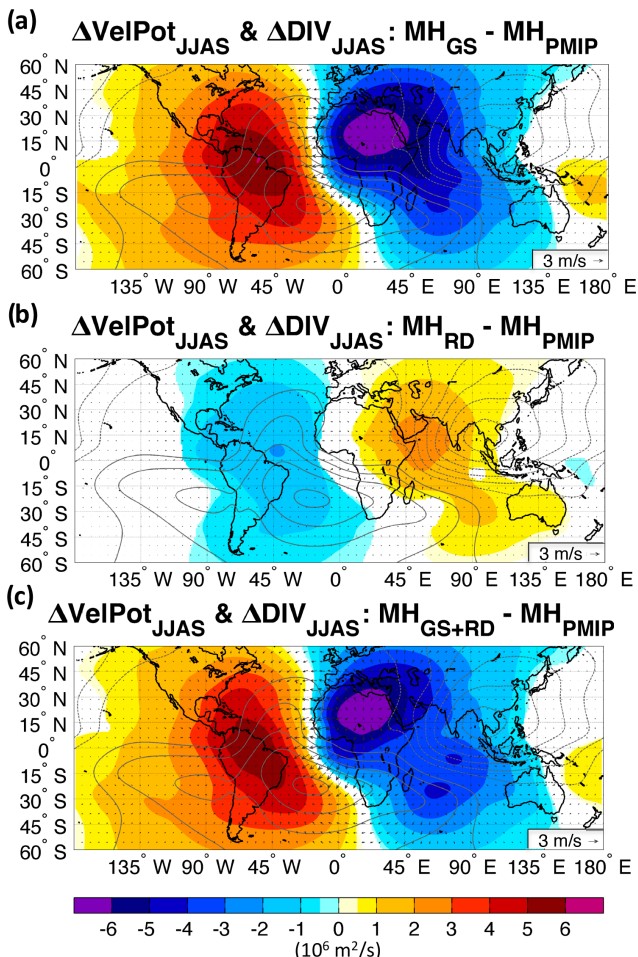

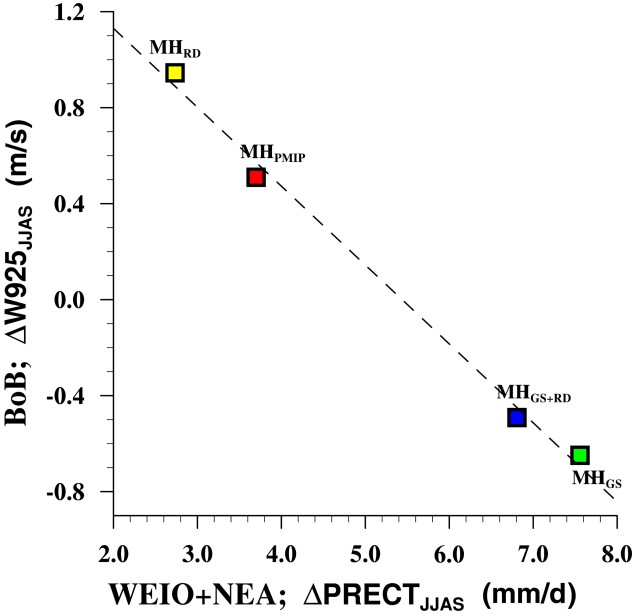

**Figure A14.** Scatter plot of summer (JJAS) changes between 925 hPa wind speed over the Bay of Bengal (BoB, 10–20° N, 85–95° E; m/s) and precipitation over the western equatorial Indian Ocean (WEIO, 5° S to 5° N, 50–65° E; mm/d) and northeastern Africa (NEA, 10–20° N, 30–45° E). Linear summation of precipitation over the two regions is considered. The changes are shown for the mid-Holocene-only orbital forcing ($MH_{PMIP}$) in red, the Sahara greening ($MH_{GS}$) in green, dust-only reduction ($MH_{RD}$) in yellow, and the Sahara greening with dust reduction ($MH_{GS+RD}$) in blue with respect to the PI reference simulation.

**Figure A13.** Changes in summer (JJAS) velocity potential (VelPot – shadings; $10^6$ m$^2$/s TS9) and divergence wind (DIV – arrows; m/s) at 200 hPa for **(a)** the Sahara greening ($MH_{GS}$); **(b)** the dust-only reduction ($MH_{RD}$); and **(c)** the Sahara greening and dust reduction ($MH_{GS+RD}$) experiments relative to the mid-Holocene-only orbital forcing ($MH_{PMIP}$). The contour lines show the climatological summer velocity potential of the $MH_{PMIP}$ experiment. Only differences significant at the 95 % confidence level using the Student $t$ test are shaded.

**Data availability.** The datasets generated and analyzed in this study are available from the corresponding author on reasonable request.

**Author contributions.** FSRP conceived the study and designed the experiments. FSRP, GM, JY, and CAJ analyzed the model output. TMM helped with the model–proxy intercomparison. All authors contributed to the interpretation of the results. FSRP and GM wrote the manuscript with contributions from all authors. TS10

**Competing interests.** The authors declare that they have no conflict of interest.

**Acknowledgements.** The authors would like to thank Qiong Zhang for sharing the global model outputs.

**Financial support.** This research has been supported by the Natural Sciences and Engineering Research Council of Canada (grant no. RGPIN-2018-04981), the Fonds de recherche du Québec – Nature et technologies (grant no. 2020-NC-268559), the Natural Environment Research Council (grant no. NE/P0067521/1) as part of the Joint Programming Initiative on Climate and the Belmont Forum for the project "Palaeo-constraints on Monsoon Evolution and Dynamics (PACMEDY)", and the Swedish Research Council (grant no. 2018-00968).

**Review statement.** This paper was edited by Qiuzhen Yin and reviewed by three anonymous referees.

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

## Remarks from the language copy-editor

CE1    Please note that, according to our house standards, all acronyms must be defined at first mention in the abstract and in the introduction. MH is defined as "mid-Holocene" here to maintain consistency with the definition given in the abstract.

CE2    Please note that this change is in accordance with our house standards and in line with the intended meaning throughout.

## Remarks from the typesetter

TS1    Please confirm affiliation.

TS2    Please give an explanation of why this needs to be changed. We have to ask the handling editor for approval. Thanks.

TS3    Please confirm the date.

TS4    Please give an explanation of why this needs to be changed. We have to ask the handling editor for approval. Thanks.

TS5    Please give an explanation of why this needs to be changed. We have to ask the handling editor for approval. Thanks.

TS6    Please give an explanation of why this needs to be changed. We have to ask the handling editor for approval. Thanks.

TS7    Please give an explanation of why this needs to be changed. We have to ask the handling editor for approval. Thanks.

TS8    Please confirm.

TS9    Please give an explanation of why this needs to be changed. We have to ask the handling editor for approval. Thanks.

TS10    Please add MAB's contributions.

TS11    Please confirm DOI.

TS12    Please provide initials of both authors.

TS13    Please provide DOI.

TS14    Please provide page range or article number.

TS15    Please provide date of last access.