# Peer review of "The remote response of the South Asian Monsoon to reduced dust emissions and Sahara greening during the middle Holocene"

_Climate of the Past, 2020_

## Referee Comment (RC1) · Anonymous Referee #1 · 18 Dec 2020

Reviewer comments#

The manuscript on "The remote response of the South Asian Monsoon to reduced dust emissions and Sahara greening during the middle Holocene" submitted by Francesco et al., attempted to show the link between the South Asian Monsoon system and the associated reduction in airborne dust concentrations related to the Sahara greening. The article is well written. The authors used climate model simulations and analyzed the data. They have carefully explained each figure by considering every minute detail observed in each figure.

Comments 1. Figure caption for figure 4 shows it is surface temperature. But in the

write-up section, it is written as sea surface temperature (SST).

2. Line 145-147 describes the precipitation dipole in the equatorial Indian Ocean. This feature resembles the precipitation pattern observed during the positive phase of the Indian Ocean Dipole (IOD). It will be interesting to explain the precipitation pattern in the context of IOD.

3. Figure 4 represents the surface temperature changes from the Pre-Industrial case. In 4a,c, the equatorial Indian ocean exhibits an IOD-like feature with cooling near the Maritime continent and warming over the western equatorial Indian Ocean. The strength of IOD, expressed in terms of IOD index, is calculated as the difference between the anomalous SST gradient over the western equatorial Indian Ocean (50E-70E and 10S-10N) and the south eastern equatorial Indian Ocean (90E-110E and 10S-0N). Figures 4a &c cooling and warming regions matches with the cooling and warming regions observed during positive IOD. Many studies have shown that many Indian summer monsoon years that accompanied positive IOD were above normal monsoon years). Even figure 4b also shows slight cooling near the Maritime continent compared to fig 4a&c. Hence it will be wise to check how the IOD- like feature might have influenced the Indian summer monsoon precipitation and under what context the IOD-like feature appears in the climate model simulations.

―――――――――――――――――――――

---

## Referee Comment (RC2) · Anonymous Referee #2 · 29 Jan 2021

Summary

Pausata et al present Earth System model simulations of the mid-Holocene which show that the greening of the Sahara and reduction in Saharan dust loading probably enhanced the South Asian monsoon during this time.

The paper is well written and extremely thorough. This paper is suitable for Climate of the Past and I would recommend publication after minor corrections as listed below.

Comments:

In a few places the reduced radiation flux is mentioned, could you calculate the

regionally-averaged the short-wave flux and include it in the text?

Lines 72-75: "Another recent study (Thompson et al, 2019) ..."

I'm don't really follow this wording. My impression was that the Thompson et al, paper showed that dust-cloud interactions had the opposite effect compare with dust-radiation interactions for rainfall in North Africa for the MH. Hopcroft & Valdes (2019) showed that dust particles radiative properties are wrong in most models and this leads to a major over-estimation of the dust-radiation effect. Perhaps you can reword this sentence to clarify.

Line 109: I think you should specify which dust optical properties are employed in the model here.

Line 448: Here I think some caveats around dust optical properties and resolved particle sizes and morphologies are needed.

Table 2: I'd be interested what the simulated surface albedo values are over the Sahara here. For example, are there other modelled processes (like wetter soil parameterisations) that could modulate the prescribed values?

Some of the references have incorrect bracketing. e.g. line 56 Pausata et al 2020, line 58, Texier et al 2000 and line 142, Zhao and Harrison. Please could you check these.

---

## Referee Comment (RC3) · Anonymous Referee #3 · 9 Mar 2021

In this paper, the authors used model experiments to evaluate the response of Indian monsoon to Sahara land cover and dust aerosol in Holocene. They found a significant remote effect of Sahara greening, which is a useful work and helps to understand the evolution of Indian monsoon under astronomical forcing. My major comments is as follows,

I wonder how and why the experiment of reduced dust is designed as in this study. From Gaetani et al 2017, it seems that the prescribed dust concentration over Sahara and Middle East is reduced by 20%. Am I right?How did they deal with the dust outside Sahara, for example, the Asian inland? Kept as PI? I understand that it is just a sensitivity run which focused on the Sahara dust effect and a prescribed 20% reduction is acceptable. But, I still wonder why they did not use a "real" dust distribution given by mid-Holocene experiment of PMIP. Does this choice of experiments affect the arguments in this paper? If the Asian dust included, does the Sahara dust intensify the Indian monsoon? This study merely considered the direct radiative effect of dust. So the simulated temperature difference is the most direct response of atmosphere. From Figure 4C, the reduced dust leads to strong warming over Middle East, Central Asia and Tibetan Plateau. It is difficult to understand such a response of surface temperature. Please explain it. Especially, the dust is simulated to cool the surface atmosphere over Middle East, which seems to contradict with most modeling studies focusing the radiative effect of Sahara dust (e.g., Albani et al., 2014). They simulated a surface warming over Sahara and Middle East because the dust is a kind of absorptive aerosol. In addition, the Sahara land cover can also warm the central Asia and Tibetan Plateau. Why? A previous modeling work (Shi et al., 2019) indicate that the radiative effect of dust can intensify the Indian monsoon, which seems opposite to this study. In that work, the dust warms the surface over Middle East, strengthens the heat low and Indian monsoon precipitation. HoweverïïjŇthis study found reduced dust could strengthen the heat low and monsoon. Does these results dependent on different models and radiation parameterization? I think the authors better give a discussion on these differences. Refs: Albani et al., Improved dust representation in the Community Atmosphere Model. JAMES, 2014. Shi et al., Snow-darkening versus direct radiative effects of mineral dust aerosol on the Indian summer monsoon onset: Role of temperature change over dust sources. ACP, 2019.

---

## Author Response (AR1)

We would like to thank all three reviewers for their comments and support to our study. We have revised the text accordingly. Below, we copy the reviewers' comments in bold and describe how each of these issues has been addressed in the revised manuscript.

**Reviewer #1**

**1. Figure caption for figure 4 shows it is surface temperature. But in the write-up section, it is written as sea surface temperature (SST).**

Thank you for pointing this out. The figure indeed shows surface temperature for both land and ocean. Hence, in the write-up section we refer to both land surface temperature and sea surface temperature. Throughout the text we have included "surface" temperature when discussing land surface temperature.

**2. Line 145-147 describes the precipitation dipole in the equatorial Indian Ocean. This feature resembles the precipitation pattern observed during the positive phase of the Indian Ocean Dipole (IOD). It will be interesting to explain the precipitation pattern in the context of IOD.**

Thank you for the comment. We have now included a discussion on the IOD-like pattern in the manuscript. Though we agree that the IOD-like pattern will have an impact on the south Asian monsoon, its effect is secondary compared to that produced by the orbital forcing. The IOD-like pattern resembles conventional reanalysis IOD primarily along the equatorial Indian ocean. There are, instead, differences in the Arabian Sea and the Bay of Bengal. A composite of positive IOD SST anomalies from ERA5 suggests that the warming along the western boundary extends into the Arabian Sea, and the Bay of Bengal has negative SST anomalies, whereas in the MH$_{PMIP}$ experiment, there is a cooling in the Arabian Sea and warming in the Bay of Bengal.

The MH forcing leads to precipitation anomalies over northeastern Africa (NEA) and the western equatorial Indian ocean (WEIO). This further induces a Matsuno-Gill-like response in the low-level winds with anomalous easterlies along the equatorial Indian ocean and a cyclonic vortex over northeastern Africa (Jalihal et al., 2019b). Thereby leading to the formation of the IOD-like pattern. The ensuing changes in the low-level jet (intensification and a northward shift) lead to a cooling in the northern Arabian Sea (through enhanced coastal upwelling) and warming in the Bay of Bengal (through reduced winds, and therefore evaporation). Thus, the SST anomalies in the Arabian Sea and the Bay of Bengal are different in the MH simulations than they are in the reanalysis data. The GS, RD, and GS+RD forcings only modulate the magnitude of the MH SST while preserving the spatial pattern.

We have now included text in the manuscript discussing the similarities and differences of this IOD-like pattern with the positive IOD as reproduced in reanalysis data.

*"A positive Indian Ocean Dipole (IOD) – like pattern develops in the Indian ocean with warmer sea surface temperatures (SSTs) of about 0.5° – 1°C over the eastern equatorial Indian Ocean up to roughly 15°N and colder anomalies of up to 1.5°C off the east coast of Indonesia (Fig. 4a). Colder SSTs are instead present over the northernmost part of the Arabian Sea, and warmer SSTs are prevalent over most of the Bay of Bengal, in contrast with the conventional IOD as seen in reanalysis data (e.g., Saji et al., 1999; Webster et al., 1999). The positive IOD-like pattern that develops under MH orbital forcing is responsible for some of the rainfall anomalies discussed above (Fig. 2a). In particular, the precipitation dipole along the equator and increased rainfall over the southern tip of India (Fig. 2a), which are typical of a positive IOD pattern (Fig. A5)."*

*"Anomalous easterlies along the equatorial Indian Ocean advect warmer water towards the western basin, leading to an increase in SSTs there. This further enhances convection over the western-equatorial Indian Ocean region. Concurrently, upwelling increases over the eastern equatorial Indian Ocean, and thus SSTs cool and precipitation decreases. As a result, a strong coupling between precipitation, circulation, and SST anomalies is established across the equatorial Indian Ocean, bearing close similarity with the pattern characteristic of the positive phase of the IOD. The subsequent changes in the low-level jet (intensification and a northward shift) lead to a cooling in the northern Arabian Sea (through enhanced coastal upwelling) and warming in the Bay of Bengal (through reduced winds, and therefore evaporation). Thus, the SST anomalies in the Arabian Sea and the Bay of Bengal are different in the MH simulations than for a positive IOD in the reanalysis data."*

**3. Figure 4 represents the surface temperature changes from the Pre-Industrial case. In 4a,c, the equatorial Indian ocean exhibits an IOD-like feature with cooling near the Maritime continent and warming over the western equatorial Indian Ocean. The strength of IOD, expressed in terms of IOD index, is calculated as the difference between the anomalous SST gradient over the western equatorial Indian Ocean (50E- 70E and 10S-10N) and the south eastern equatorial Indian Ocean (90E-110E and 10S-0N). Figures 4a &c cooling and warming regions matches with the cooling and warming regions observed during positive IOD. Many studies have shown that many Indian summer monsoon years that accompanied positive IOD were above normal monsoon years). Even figure 4b also shows slight cooling near the Maritime continent compared to fig 4a&c. Hence it will be wise to check how the IOD-like feature might have influenced the Indian summer monsoon precipitation and under what context the IOD-like feature appears in the climate model simulations.**

As mentioned in the reply to the previous comment, the IOD-like pattern in our model is primarily along the equatorial Indian Ocean. The precipitation anomalies off the equator produced by a positive IOD are also different, particularly over India and the Bay of Bengal, compared to the anomalies in the simulations. In the MH$_{PMIP}$ experiment, there is a reduction in precipitation over

most of the Bay of the Bengal, and an increase in precipitation over India (Fig. 2). On the other hand, a positive IOD produces an increase in precipitation in the northern Bay of Bengal and the core monsoon zone and a decrease in the southern Bay of Bengal, peninsular India, and the Himalayan foothills.

To understand the impact of the IOD-like pattern on precipitation over India, we have scattered the IOD-index from the four experiments ($MH_{PMIP}$, $MH_{GS}$, $MH_{RD}$, $MH_{GS+RD}$) with the anomalies in precipitation over India.
We find that the IOD index is inversely related to the precipitation over India in these experiments. Thus, further highlighting that the impact of other forcings dominates over that of the IOD (Fig. A7).

We have included the following sentences in the manuscript:

*"There are, however, differences in the anomalies over India and the Bay of Bengal in the simulations since the orbital forcing primarily drives these anomalies. A classical positive IOD leads to an increase in precipitation over the core monsoon zone and the northern Bay of Bengal and a decrease in precipitation over the southern Bay of Bengal, peninsular India, and the Himalayan foothills (see Saji et al., 1999; Ashok et al., 2001). The orbital forcing leads to a different response over land and the ocean (Fig. 2a)."*

*"The positive IOD-like pattern is still present, particularly when considering relative anomalies as the SSTs over the equatorial Indian Ocean are generally warmer compared to the $MH_{PMIP}$. The IOD-index is inversely related to the Indian monsoon rainfall (Fig. A7). Thus, suggesting that the orbital, vegetation and dust forcings have a dominant effect on the SAM."*

**Reviewer #2**

**1. In a few places the reduced radiation flux is mentioned, could you calculate the regionally-averaged the short-wave flux and include it in the text?**

We have included a new figure (Fig. 5) in which we show the change in shortwave net radiation at the top of the atmosphere (Fig. R1). There is enhanced shortwave radiation across the bulk of the domain relative to the PI in all simulations. We comment further on this figure in the main text in Sect. 3.1 Surface Temperature.

[Figure]

**Figure R1.** Changes in summer (JJAS, June to September) top of the atmosphere shortwave radiation (RAD$^{TOP}$; W/m²) for the (a) middle Holocene only orbital forcing (MH$_{PMIP}$); (b) the Sahara greening (MH$_{GS}$); (c) the only dust reduction (MH$_{RD}$); and (d) the Sahara greening and dust reduction (MH$_{GS+RD}$) experiments relative to the pre-industrial (PI) reference simulation. The contour lines follow the colorbar scale (the 0 lines are omitted for clarity). Only differences significant at the 95% confidence level using the Student $t$ test are shaded.

**2. Lines 72-75: "Another recent study (Thompson et al, 2019) ..." I'm don't really follow this wording. My impression was that the Thompson et al, paper showed that dust-cloud interactions had the opposite effect compare**

**with dust-radiation interactions for rainfall in North Africa for the MH. Hopcroft & Valdes (2019) showed that dust particles radiative properties are wrong in most models and this leads to a major over-estimation of the dust-radiation effect. Perhaps you can reword this sentence to clarify.**

We have changed it to read:

*"Another recent study (Thompson et al., 2019) has suggested a contribution from dust aerosol reduction of about 15-20% to the total rainfall over the Sahara; however, they also revealed that dust-cloud interactions have the opposite effect compared to the direct radiative effect on rainfall in northern Africa during the MH. Hopcroft and Valdes (2019) show the dependence on the modelled dust optical properties and particle size range of the impacts on WAM rainfall, leading to potential overestimation of the direct radiative effect on precipitation."*

**3. Line 109: I think you should specify which dust optical properties are employed in the model here.**

We have included the following text in section 2 as well as table A1 that summarize the properties:

*"Relevant for this study, vegetation cover and monthly aerosol concentrations (Tegen et al., 1997) are prescribed in the model; however, the indirect effect of aerosols on clouds is not considered. A detailed description of the aerosol components can be found in Hess et al. (1998). The main characteristics of dust particles are reported in Table A1."*

**Table A1:** Aerosol optical depth, single scattering albedo and composition of the mineral dust for a relative humidity of 50%.

| Type | RH (%) | AOD at 550 nm | SSA ($\omega_0$) | Component | Number (cm$^{-3}$) | Mass ($\mu g/m^3$) |
|---|---|---|---|---|---|---|
| "Desert" dust-like | 50 | 0.037 | 0.888 | Total | 2300 | 225.8 |
| | | | | Water soluble | 2000 | 4.0 |
| | | | | Mineral (nuclei) | 269.5 | 7.5 |
| | | | | Mineral (accum.) | 30.5 | 168.7 |
| | | | | Mineral (coarse) | 0.142 | 45.6 |

**4. Line 448: Here I think some caveats around dust optical properties and resolved particle sizes and morphologies are needed.**

We have included the following discussion:

*"However, the simulated impact of dust changes needs further investigation, as rainfall in tropical regions is strongly affected by the specific prescription of dust optical properties. In particular, the choice of the single scattering albedo $\omega_0$ can significantly alter the effect of dust on precipitation via the so-called "heat pump" effect (Lau et al., 2009). The atmospheric dust layer in which the dust particles are moderately-to-highly absorbing (single scattering albedo $\omega_0 < 0.95$) warms the atmosphere enhancing deep convection and intensifying the monsoonal precipitation (Lau et al., 2009). In particular, EC-Earth has a single scattering albedo of 0.89 at 550 nm (Table A1). Such a value is too absorbing compared to observations (see figure 1 in Albani et al., 2014) and consequently the radiative impact of dust is likely overestimated as also pointed out in Hopcroft and Valdes (2019). Furthermore, Albani and Mahowald (2019) showed how different choices in terms of dust optical properties and size distributions may yield opposite results in terms of rainfall changes. For example, Shi et al. (2019) showed that the dust radiative effect intensifies the SAM, which is opposite to our results. This difference results in a warming of the Tibetan Plateau and Central Asia when reducing dust under a Green Sahara, likely associated with a decrease in precipitation in the region.*

*A further caveat of our work with respect to dust is that we rely on an idealised dust reduction pattern, as opposed to more realistic global dust modulation patterns (e.g. Albani et al., 2015). However, in the EC-Earth simulations most of the changes in the WAM intensity and the teleconnection to the SAM were associated to changes in surface albedo due to greening of the Sahara. The surface albedo changes were then further enhanced by dust reduction. This rainfall response in the WAM is opposite to what one would expect from a reduced "heat pump" effect (decreased rainfall), suggesting that the "heat pump" effect is overwhelmed by the changes in surface albedo under green Sahara conditions in EC-Earth simulations. Moreover, previous work (Gaetani et al., 2017), has shown that the results from different dust distributions are very similar, and do not alter the qualitative conclusion that dust changes amplify the effects associated with land surface changes."*

**5. Table 2: I'd be interested what the simulated surface albedo values are over the Sahara here. For example, are there other modelled processes (like wetter soil parameterisations) that could modulate the prescribed values?**

No, the albedo of the surface is fixed and not modulated by modeled processes. We now specify this explicitly in the text, and provide albedo values in Table 2.

**6. Some of the references have incorrect bracketing. e.g. line 56 Pausata et al 2020, line 58, Texier et al 2000 and line 142, Zhao and Harrison. Please could you check these.**

Thank you for pointing them out. We have fixed them.

**Reviewer #3**
**1. My major comments is as follows, I wonder how and why the experiment of reduced dust is designed as in this study. From Gaetani et al 2017, it seems that the prescribed dust concentration over Sahara and Middle East is reduced by 20%. Am I right? How did they deal with the dust outside Sahara, for example, the Asian inland? Kept as PI? I understand that it is just a sensitivity run which focused on the Sahara dust effect and a prescribed 20% reduction is acceptable. But, I still wonder why they did not use a "real" dust distribution given by mid-Holocene experiment of PMIP. Does this choice of experiments affect the arguments in this paper? If the Asian dust included, does the Sahara dust intensify the Indian monsoon?**

Thanks for pointing this out. We realized that in the previous version of the manuscript not enough information was given to the reader regarding the change in dust. We have also included the main characteristics of dust particles in our model in Table A1.
In Section 2 we have modified/included the following paragraph:

*"In the MH$_{RD}$ ('Reduced Dust') setup, the dust concentration over northern Africa is reduced by up to 80% relative to pre-industrial values (see Figs. 1 and S1 in Gaetani et al., 2017). Outside northern Africa, dust concentrations smoothly transition to pre-industrial values. Over India and the Arabian Sea the reduction of dust concentrations ranges between 20% (Eastern Indian subcontinent) and 60% (Horn of Africa and Middle East); for more details see figure S1 in Pausata et al. (2016)."*

In PMIP4, sensitivity experiments with changes in dust following Albani et al. 2015, are indeed present (Otto-Blienser et al., 2017). However, we have not included them as we had already performed the experiments at that time.
A different dust distribution may lead to slightly different results, but it will unlikely lead to significantly different outcomes. For example, in Gaetani et al. 2017 the results from two different dust distributions are presented and the results were very similar. Furthermore, the dust changes in our model simply amplify the effect associated with land surface changes.
We have included the following short discussion in the manuscript that reads:

*"A further caveat of our work with respect to dust is that we rely on an idealised dust reduction pattern, as opposed to more realistic global dust modulation patterns (e.g. Albani et al., 2015). However, in the EC-Earth simulations most of the changes in the WAM intensity and the teleconnection to the SAM were associated to changes in surface albedo due to greening of the Sahara. The surface albedo changes were then further enhanced by dust reduction. This rainfall response in the WAM is opposite to what one would expect from a reduced "heat pump" effect (decreased rainfall), suggesting that the "heat pump" effect is overwhelmed by the changes in surface albedo under green Sahara conditions in EC-Earth simulations. Moreover, previous work (Gaetani et al., 2017), has shown that the results from different dust distributions are very similar, and do not alter the qualitative*

*conclusion that dust changes amplify the effects associated with land surface changes."*

**2. From Figure 4C, the reduced dust leads to strong warming over Middle East, Central Asia and Tibetan Plateau. It is difficult to understand such a response of surface temperature. Please explain it. Especially, the dust is simulated to cool the surface atmosphere over Middle East, which seems to contradict with most modeling studies focusing the radiative effect of Sahara dust (e.g., Albani et al., 2014). They simulated a surface warming over Sahara and Middle East because the dust is a kind of absorptive aerosol.**

The effect of dust on surface temperature is related to the surface albedo and changes in rainfall. In particular, a decrease in rainfall and hence cloud cover over southern Arabian Peninsula and southern India favors a surface warming. Furthermore, while a reduction of dust does cause a cooling in the mid-troposphere, it allows more radiation to reach the surface and hence favors a surface warming in our model.

The paragraph discussing figure 4c now reads:

*"Reduced Saharan dust ($MH_{RD}$) leads to a widespread surface warming over the Arabian Peninsula, the Arabian Sea, and the Indian subcontinent (cf. panels a and c in Figure 4 and see also Figure A5b). Such warming is partially due to a reduction in rainfall (Fig. A3b) and hence cloud cover in particular over southern India and southern Arabic Peninsula. Furthermore, the reduced dust layer, while it leads to a decrease in temperature in the mid-troposphere as dust is moderately-to-highly absorbing (single scattering albedo $\omega_0 < 0.95$, see Table A1), increases the incoming solar radiation reaching the surface and hence favours surface warming. For the same reason, the cold SST anomalies in the northernmost Arabian Sea in the $MH_{PMIP}$ experiment are replaced by a modest warm anomaly."*

**3. In addition, the Sahara land cover can also warm the central Asia and Tibetan Plateau. Why? A previous modeling work (Shi et al., 2019) indicate that the radiative effect of dust can intensify the Indian monsoon, which seems opposite to this study. In that work, the dust warms the surface over Middle East, strengthens the heat low and Indian monsoon precipitation. However, this study found reduce dust could strengthen the heat low and monsoon. Does these results dependent on different models and radiation parameterization? I think the authors better give a discussion on these differences.**

Our experiments do show a warming in central Asia and Tibetan Plateau. This warming is most likely associated with the reduction in precipitation over the Plateau. In fact, the $MH_{RD}$ does not show any changes in precipitation relative to $MH_{PMIP}$ and no changes in temperature, while the changes in precipitation are larger in $MH_{GS+RD}$ relative to $MH_{GS}$ and so is the warming.

We have included the following paragraphs in the discussion section:

*"However, the simulated impact of dust changes needs further investigation, as rainfall in tropical regions is strongly affected by the specific prescription of dust optical properties. In particular, the choice of the single scattering albedo $\omega_0$ can significantly alter the effect of dust on precipitation via the so-called "heat pump" effect (Lau et al., 2009). The atmospheric dust layer in which the dust particles are moderately-to-highly absorbing (single scattering albedo $\omega_0 < 0.95$) warms the atmosphere enhancing deep convection and intensifying the monsoonal precipitation (Lau et al., 2009). In particular, EC-Earth has a single scattering albedo of 0.89 at 550 nm (Table A1). Such a value is too absorbing compared to observations (see figure 1 in Albani et al., 2014) and consequently the radiative impact of dust is likely overestimated as also pointed out in Hopcroft and Valdes (2019). Furthermore, Albani and Mahowald (2019) showed how different choices in terms of dust optical properties and size distributions may yield opposite results in terms of rainfall changes. For example, Shi et al. (2019) showed that the dust radiative effect intensifies the SAM, which is opposite to our results. This difference results in a warming of the Tibetan Plateau and Central Asia when reducing dust under a Green Sahara, likely associated with a decrease in precipitation in the region."*